# Sensory TRP channels contribute differentially to skin inflammation and persistent itch

Jing Feng[1], Pu Yang[1], Madison R. Mack[2], Dariia Dryn[1,3], Jialie Luo[1], Xuan Gong[1,4], Shenbin Liu[1], Landon K. Oetjen[2], Alexander V. Zholos[3], Zhinan Mei[4], Shijin Yin[4], Brian S. Kim[2] & Hongzhen Hu[1]

Although both persistent itch and inflammation are commonly associated with allergic contact dermatitis (ACD), it is not known if they are mediated by shared or distinct signaling pathways. Here we show that both TRPA1 and TRPV1 channels are required for generating spontaneous scratching in a mouse model of ACD induced by squaric acid dibutylester (SADBE), a small molecule hapten, through directly promoting the excitability of pruriceptors. TRPV1 but not TRPA1 channels protect the skin inflammation, as genetic ablation of TRPV1 function or pharmacological ablation of TRPV1-positive sensory nerves promotes cutaneous inflammation in the SADBE-induced ACD. Our results demonstrate that persistent itch and inflammation are mediated by distinct cellular and molecular mechanisms in a mouse model of ACD. Identification of distinct roles of TRPA1 and TRPV1 in regulating itch and inflammation may provide new insights into the pathophysiology and treatment of chronic itch and inflammation in ACD patients.

---

[1] Department of Anesthesiology, The Center for the Study of Itch, Washington University School of Medicine, St. Louis, MO 63110, USA. [2] Division of Dermatology, Department of Medicine, The Center for the Study of Itch, Washington University School of Medicine, St. Louis, MO 63110, USA. [3] Department of Biophysics, Institute of Biology, Taras Shevchenko National University of Kyiv, Kyiv 01601, Ukraine. [4] College of Pharmacy, South-Central University for Nationalities, Wuhan 430074, China. Jing Feng and Pu Yang contributed equally to this work. Correspondence and requests for materials should be addressed to H.H. (email: hongzhen.hu@wustl.edu)

Contact hypersensitivity (CHS) in allergic contact derma-titis (ACD) is a cutaneous immune response elicited by hapten sensitization, which is primarily mediated by der-mal dendritic cells (DC) and both CD4[+] and CD8[+] T cells [1–3]. Squaric acid dibutylester (SADBE), a small molecule hapten, is commonly used in the treatment of alopecia areata and warts but often causes CHS[4, 5]. Consistent with clinical observations, SADBE can also effectively induce CHS in a mouse model of ACD in which pre-sensitization and subsequent challenges with SADBE produce marked epidermal hyperplasia and spontaneous scratching behaviors, recapitulating symptoms of ACD in humans[6]. However, the molecules and cells involved in the generation of SADBE-induced CHS, especially persistent itch, remain elusive.

Mammalian transient receptor potential (TRP) channels are a group of cation channels expressed by numerous animal cells and play important roles in both health and disease[7]. Several TRP channels, especially the TRPA1 and TRPV1 channels, are selec-tively expressed by a subpopulation of small-diameter primary sensory neurons and serve as molecular sensors for both innoc-uous and noxious thermal, mechanical, and chemical cues in our environment[8–12]. TRPA1 and TRPV1 are involved in the pathogenesis of both inflammatory and neuropathic pain and also serve as key downstream signaling molecules for histamine-independent and histamine-dependent itch, respectively[13, 14]. Recent studies have also shown that enhanced excitability of primary sensory neurons expressing itch-mediating mas-rela-ted G protein-coupled receptors MRGPRA3 and MRGPRD and several chemokines and chemokine receptors were correlated with sensory hypersensitivity in response to pre-sensitization and subsequent challenges with SADBE in mice[6, 15], thereby provid-ing a cellular mechanism accounting for spontaneous scratching in this mouse model of ACD. Interestingly, activation of TRPA1 and TRPV1 channels could either promote or suppress skin inflammation in CHS in addition to their roles in pain and itch signal transduction[16]. For instance, skin edema was significantly diminished in TRPA1-deficient mice while TRPV1 activation inhibited the development of inflammation in a mouse model of oxazolone-induced CHS[17, 18]. Although it is well known that both TRPA1 and TRPV1 are major players in the generation of pain, itch, and inflammation, it is not clear if these sensory TRP channels are involved in regulating skin inflammation and per-sistent itch in the SADBE-induced ACD model.

Here we show that SADBE is an activator of both TRPA1 and TRPV1 by directly interacting with specific amino acid residues located at their intracellular protein domains. Both sensory TRP channels are required for generating the SADBE-induced per-sistent itch. On the other hand, the TRPV1 channels exert a protective role in suppressing SADBE-induced ear edema through modulating the function of dermal macrophages. Our results suggest that chronic skin inflammation and persistent itch in the SADBE-induced CHS are mediated by distinct molecular mechanisms that rely on the functions of sensory TRP channels.

## Results

**Lmphocytes contribute to skin inflammation but not itch**. Hapten-induced CHS in ACD is commonly accepted as a result from activated adaptive immune response[19]. Although the exact mechanism is not fully understood, conventional protocols comprising pre-sensitization and subsequent elicitation of defined skin areas with SADBE are frequently used for the treatment of alopecia areata and warts. In the pre-sensitization phase, naive T cells are activated by antigen-presenting cells in the draining Lymph nodes. After exposure to the same hapten in the elicitation phase, antigen-specific T cells initiate the inflammatory process

resulting in skin inflammation[19]. Since pre-sensitization is critical to the induction of T cell-mediated immunity in ACD, we investigated the effect of pre-sensitization on SADBE-induced inflammation and spontaneous scratching. We measured the ear thickness and spontaneous scratching in wild-type (wt) *C57BL/6J* mice pre-treated with either vehicle acetone or SADBE for 3 days and challenged with SADBE 5 days later as described[6] (Fig. 1a). Although the group receiving SADBE for both pre-sensitization and subsequent elicitation displayed a much severer inflamma-tory reaction compared with the group pre-treated with acetone and then challenged with SADBE, the latter group also showed a significantly increased ear thickness when compared with the group without SADBE challenges (Fig. 1e). These results suggest that SADBE can evoke both lymphocyte-dependent and lymphocyte-independent inflammatory responses, similar to those seen in ACD and irritant contact dermatitis (ICD), which is a nonallergic inflammatory reaction[20]. Interestingly, the mice pre-treated with either acetone or SADBE developed comparable spontaneous scratching as long as they were later challenged with SADBE (Fig. 1i). Therefore, T cell-mediated immunity con-tributes partly to SADBE-induced skin inflammation but not persistent itch.

To further investigate whether T cells contribute differentially to SADBE-induced skin inflammation and persistent itch, we measured both skin inflammation and spontaneous scratching produced by SADBE treatment in the recombination activating gene 1 knockout ($Rag1^{-/-}$) mice, which lack both mature T cells and B cells[21] (Fig. 1b). Surprisingly, both ear thickness and the number of spontaneous scratches after SADBE challenges were not significantly different between the wt and $Rag1^{-/-}$ mice (Fig. 1f, j). Moreover, wt mice treated with FTY720, a sphingosine 1-phosphate (S1P) analog that blocks egress of T cells from lymph nodes and subsequent generation of primary effector T cells[22], did not affect SADBE-induced skin inflammation and spontaneous scratching (Supplementary Fig. 1). Interestingly, it was reported that besides T cells and B cells the natural killer (NK) cells were also required for generating the CHS in ACD induced by either 2,4-dinitrofluorobenzene or oxazolone[23]. Thus, we determined the function of NK cells in SADBE-induced ACD by using antibody-mediated depletion as described[23]. Consistent with previous report, NK cell depletion in wt mice didn't alleviate SADBE-induced inflammation and persistent itch (Fig. 1c, g and k). Combined, these results suggest that abolition of T cell or NK cell priming alone is not sufficient to affect SADBE-induced skin inflammation and persistent itch.

To achieve a complete deficiency in lymphocytes, we used antibody-mediated depletion of NK cells in the $Rag1^{-/-}$ mice (Fig. 1d). Indeed, skin inflammation induced by SADBE in the $Rag1^{-/-}$ mice treated with anti-NK1.1 antibodies was signifi-cantly reduced compared with mice receiving control IgG antibodies (Fig. 1h). Surprisingly, the number of SADBE-induced spontaneous scratches was still not significantly different between the $Rag1^{-/-}$ mice with and without receiving the anti-NK1.1 antibodies (Fig. 1l), suggesting that the SADBE-induced spontaneous scratching does not require lymphocyte-mediated immunity.

**TRPV1 and TRPA1 are required for SADBE-induced chronic itch**. The finding that the persistent itch induced by SADBE challenges does not rely on adaptive immunity suggests the involvement of alternative mechanisms. TRP channels, especially TRPA1 and TRPV1, are critically involved in generating both acute and chronic itch and often serve as downstream signaling effectors converging many signaling cascades resulting from activation of numerous G protein-coupled receptors in the itch

transduction pathways[7, 24]. We therefore investigated whether these two sensory ion channels are involved in SADBE-induced persistent itch. Since both TRPA1 and TRPV1 are co-expressed by the TRPV1-expressing primary afferent C-fibers[25, 26] which are essential to itch responses induced by various pruritogens and chemical irritants[12, 27], we first tested if pharmacological ablation of the TRPV1-positive C-fibers with the ultrapotent capsaicin analog resiniferotoxin (RTX)[28] affects the SADBE-induced spontaneous scratching (Fig. 2a). Strikingly, the number of spontaneous scratches after the SADBE challenges was markedly reduced in mice treated with RTX when compared with mice treated with vehicle only (Fig. 2c), suggesting that the TRPV1-positive C-fibers are indeed required for generating the SADBE-induced spontaneous scratching. We next measured the numbers of spontaneous scratches in the $Trpa1^{-/-}$, $Trpv1^{-/-}$ and $Trpa1^{-/-}/Trpv1^{-/-}$ double knockout (dKO) mice to ask if these two sensory TRP channels are involved in the generation of SADBE-induced spontaneous scratching (Fig. 2b). Remarkably, the numbers of spontaneous scratches were substantially reduced in all three groups compared with their wt controls (Fig. 2d). Furthermore, SADBE-induced persistent itch was significantly suppressed by pharmacological inhibition of TRPA1 and/or TRPV1 function with specific antagonists (Supplementary Fig 2a). Taken together, these results suggest that both TRPA1 and TRPV1 are required for generating the SADBE-induced persistent itch.

**TRPA1 and TRPV1 mediate SADBE-induced activation of DRG.** There are at least two possibilities for the involvement of TRPA1 and TRPV1 channels in the generation of SADBE-induced persistent itch. One possibility is that SADBE could stimulate release of endogenous pruritogens from resident skin cells except for lymphocytes, for instance, keratinocytes, mast cells, or innate immune cells, which subsequently activate the TRPV1 and/or TRPA1 channels expressed by pruriceptors[7, 29]. Alternatively, SADBE could act as a chemical irritant that penetrates the compromised skin barrier resulting from repeated SADBE challenges and directly promotes excitability of the cutaneous pruriceptors through activating the TRPA1 and/or TRPV1 channels. We tested this possibility by using live-cell $Ca^{2+}$ imaging on dissociated mouse DRG neurons. Bath application of 3 mM SADBE could indeed evoke a large intracellular $Ca^{2+}$ response in about 56.7% of the wt DRG neurons (Fig. 3a and i). Interestingly, SADBE elicited an intracellular $Ca^{2+}$ response in comparable proportions of $Trpa1^{-/-}$ or $Trpv1^{-/-}$ DRG neurons where allyl isothiocyanate (AITC)- or capsaicin-elicited $Ca^{2+}$ response was absent, respectively (Fig. 3b, c and i). On the other hand, SADBE-induced $Ca^{2+}$ response was completely absent from the DRG neurons isolated from the $Trpa1^{-/-}/Trpv1^{-/-}$ dKO mice (Fig. 3d and i). Taken together, these results suggest that both TRPA1 and TRPV1 are sensitive to SADBE and only genetic ablation of both TRPA1 and TRPV1 functions is able to abolish the SADBE-elicited $Ca^{2+}$ response in DRG neurons by eliminating the mutual compensatory effect observed in the DRG neurons isolated from the single $Trpa1^{-/-}$ or $Trpv1^{-/-}$ mice.

To measure the direct effect of SADBE on the excitability of primary sensory neurons we performed whole-cell patch clamp recordings on dissociated DRG neurons. Under the current clamp mode, bath application of 3 mM SADBE evoked a large

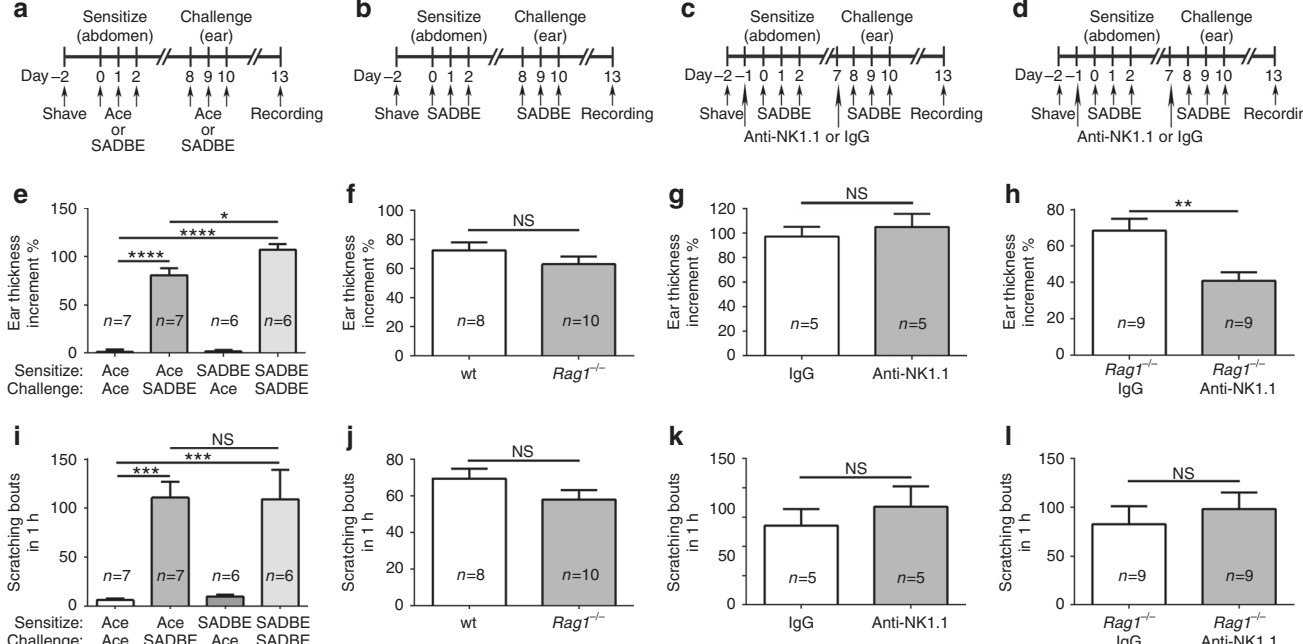

**Fig. 1** T lymphocytes contribute partly to skin inflammation but not persistent itch. **a**, **e**, **i** Schematic experimental protocol in **a** was used to test the effect of pretreatment with vehicle (acetone) or SADBE on ear thickness increment (**e**) and spontaneous scratching (**i**) induced by subsequent elicitation of SADBE. Data are presented as mean ± SEM. NS, not significant. *Asterisks* indicate statistical significance. *$p < 0.05$, ***$p < 0.001$, ****$p < 0.0001$, ANOVA; **b**, **f**, **j** Pre-sensitization and subsequent elicitation with SADBE (schematic protocol in **b**) induced comparable ear thickness increment (**f**) and spontaneous scratching (**j**) between wt and $Rag1^{-/-}$ mice. Data are presented as mean ± SEM. NS, not significant, Student's *t*-test; **c**, **g**, **k** Pre-sensitization and subsequent elicitation with SADBE (schematic protocol in **c**) elicited comparable ear thickness increment (**g**) and spontaneous scratching (**k**) between wt mice treated with either control mouse IgGs or anti-NK1.1. Data are presented as mean ± SEM. NS, not significant, Student's *t*-test; **d**, **h**, **l** The $Rag1^{-/-}$ mice received injections of anti-NK1.1 antibodies had significantly less ear thickness increment vs. $Rag1^{-/-}$ mice receiving only control mouse IgGs (**h**) after pre-sensitization and subsequent elicitation with SADBE (schematic protocol in **d**) but the spontaneous scratching behaviors were comparable between $Rag1^{-/-}$ mice receiving anti-NK1.1 and control mouse IgGs (**l**). Data are presented as mean ± SEM. NS, not significant. *Asterisks* indicate statistical significance. **$p < 0.01$, Student's *t*-test

membrane depolarization and increased the number of action potential firing in wt DRG neurons which also responded to both AITC and capsaicin (Fig. 3e and j). Consistent with the $Ca^{2+}$ imaging results, SADBE evoked comparable membrane depolarizing responses in the DRG neurons isolated from the $Trpa1^{-/-}$, $Trpv1^{-/-}$ or wt mice (Fig. 3f, g and j). However, no membrane depolarization was observed in response to SADBE, AITC or capsaicin in the DRG neurons isolated from the $Trpa1^{-/-}/Trpv1^{-/-}$ dKO mice (Figs. 3h and j). To further confirm that TRPA1 and TRPV1 were the only receptors activated by SADBE in DRG neurons, we used a higher concentration (10 mM) of SADBE to fully activate DRG neurons. Although 10 mM SADBE elicited a robust membrane depolarization associated with spontaneous action potential firings isolated from wt mice and blunted the AITC and capsaicin responses in the same DRG neurons (Supplementary Fig. 3a, 3b and 3e), no membrane depolarization, action potential firing or $Ca^{2+}$ influx were evoked by 10 mM SADBE in DRG neurons isolated from $Trpa1^{-/-}/Trpv1^{-/-}$ dKO mice in both current clamp recordings and $Ca^{2+}$ imaging assays (Supplementary Fig. 3c, 3d, 3f and 3g). These results suggest that SADBE could directly activate the primary sensory neurons and TRPA1 and TRPV1 are the indispensable targets for SADBE in these neurons.

**SADBE directly activates recombinant TRPA1 and TRPV1.** To further test the possibility that SABDE directly activates both TRPA1 and TRPV1 channels we transfected HEK293 cells with either human TRPA1 or mouse TRPV1 construct and measured TRP channel activities using live-cell $Ca^{2+}$ imaging and whole-cell patch clamp recordings. As predicted, SADBE evoked large intracellular $Ca^{2+}$ responses in both TRPA1- and TRPV1-transfected HEK293 cells (Fig. 4a and b). In marked contrast, HEK293 cells transfected with mouse TRPV3, rat TRPV4, or vector control didn't show detectable intracellular $Ca^{2+}$ responses to SADBE (Supplementary Fig. 4). Furthermore, SADBE activated membrane currents in HEK293 cells transfected with either TRPA1 or TRPV1 construct but not the vector control under

voltage-clamp configuration (Fig. 4c–f). The effect of SADBE was concentration-dependent with an $EC_{50}$ of 1.30 and 7.26 mM for HEK293 cells transfected with TRPA1 or TRPV1, respectively (Figs. 4g and h, Supplementary Tables 1 and 2).

We next determined the amino acid residues in TRPA1 protein that confer SADBE sensitivity by generating both cysteine and lysine mutants that are known to disrupt activation of TRPA1 by noxious electrophilic chemical reagents using site-directed mutagenesis[30], [31]. The $EC_{50}$ value for SADBE activation of TRPA1 had a 5-fold increase in the TRPA1-K710A mutant in comparison to wild-type TRPA1 (Fig. 4g, Supplementary Table 1). However, TRPA1-3C (a combination of C621S, C641S, and C665S) mutant barely displayed any sensitivity to SADBE while retained significantly larger responses to flufenamic acid (FFA), a non-electrophilic TRPA1 activator (Supplementary Fig. 5), suggesting that SADBE uses the same interaction sites for electrophiles to activate TRPA1 (Fig. 4g, Supplementary Table 1). We also made TRPV1 mutants in which the capsaicin interaction site was disrupted[32]. Among these mutants, the potency of SADBE activation of TRPV1 was not significantly affected by substituting TRPV1-R115 and even increased in TRPV1-M548 but the SADBE-elicited current was almost abolished in the TRPV1-Y512, TRPV1-S513 and TRPV1-T551 mutants (Fig. 4h, Supplementary Table 2), suggesting that SADBE interacts with TRPV1 through the same interaction sites for capsaicin activation of the channel. Combined, our results provide strong evidence that SADBE is an activator for both TRPV1 and TRPA1 channels.

**SADBE elicits a TRPA1-dependent acute scratching behavior.** Given that SADBE is an activator of both TRPA1 and TRPV1, we asked if subcutaneous application of SADBE could elicit an acute scratching response similar to that evoked by chloroquine and histamine. First, we employed the "cheek assay", which reportedly distinguished itch and pain behaviors by measuring hind paw scratching and forelimb wiping, respectively[33]. When injected into the cheeks of the wt mice, 30 mM SADBE produced a robust scratching response without any detectable wiping behavior

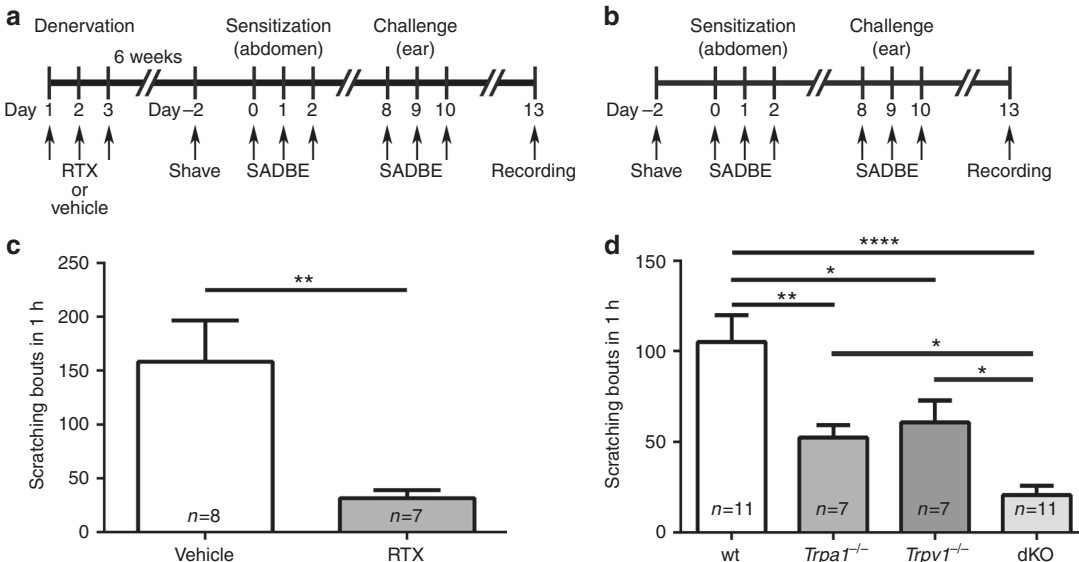

**Fig. 2** Both TRPA1 and TRPV1 channels are involved in SADBE-induced persistent itch. **a** Schematic protocol of nociceptor ablation and induction of SADBE-induced ACD model; (**b**) Schematic protocol of SADBE-induced ACD model; (**c**) SADBE-induced spontaneous scratching in wt mice pre-treated with vehicle or RTX. Data are presented as mean ± SEM. Asterisks indicate statistical significance. **p < 0.01, Student's t-test; (**d**) SADBE-induced spontaneous scratching in wt, $Trpa1^{-/-}$, $Trpv1^{-/-}$, and $Trpa1^{-/-}/Trpv1^{-/-}$ dKO mice. Data are presented as mean ± SEM. Asterisks indicate statistical significance. *p < 0.05; **p < 0.01; ****p < 0.0001, ANOVA

(Fig. 5a). Furthermore, we didn't observe any paw licking or flicking behavior within the first 10 mins after intraplantar injections of 30 mM SADBE (Supplementary Fig. 6a). To further exclude the possibility that SADBE contributes to mechanical and thermal pain sensitivities, we performed Von Frey and Hargreaves tests after paw injections of 30 mM SADBE and no significant difference was found in paw withdrawal threshold to mechanical stimuli or paw withdrawal latency to heat stimuli compared with mice injected with vehicle only (Supplementary Fig. 6b, c). These results suggest that SADBE acts primarily as an itch-evoking compound when applied acutely at the given dosage. We next tested if TRPA1 and/or TRPV1 were involved in acute scratching in mice subjected to intradermal injections of SADBE. As expected, SADBE but not vehicle control produced a robust scratching behavior in wt mice but not in the $Trpa1^{-/-}/Trpv1^{-/-}$ dKO mice (Fig. 5b), suggesting that TRPA1 and/or TRPV1 are essential to SADBE-induced acute itch response. Surprisingly, although the $Trpa1^{-/-}$ mice displayed a significantly attenuated scratching response, which is similar to that observed in the $Trpa1^{-/-}/Trpv1^{-/-}$ dKO mice, the SADBE-elicited scratching was not significantly reduced in the $Trpv1^{-/-}$ mice (Fig. 5b). These results suggest that TRPA1 is the predominant itch receptor mediating the acute scratching response induced by subcutaneous applications of SADBE, although both TRPA1 and TRPV1 are involved in generating persistent itch in the SADBE-induced ACD model.

**TRPV1 deficiency promotes skin inflammation**. When activated, the cutaneous primary sensory nerve endings, especially the TRPV1-expressing nociceptive C-fibers, mediate both pain (afferent function) and neurogenic inflammation (efferent function) and may also contribute to allergic inflammatory responses[34, 35]. Further, previous studies showed that RTX-mediated ablation of the TRPV1-positive sensory fibers reduces imiquimod (IMQ)-induced psoriasis-like ear skin inflammation resulting from activation of the IL-23/IL-17 pathway in the dermal dendritic cells/γδ T cells[28]. To investigate if TRPV1-expressing C-fibers are involved in SADBE-induced ear edema, we pre-treated the mice with systemic administration of RTX to ablate the TRPV1-positive sensory nerve endings. Surprisingly, the ear edema induced by SADBE challenges was profoundly increased in the RTX-treated mice (Figs 6a and c), suggesting that the TRPV1-expressing sensory nerve endings play a protective role in SADBE-induced skin inflammation. Therefore, the TRPV1-expressing sensory fibers can differentially regulate skin inflammation in an etiology-specific manner.

To clarify if TRPA1 and/or TRPV1 channels mediate this protective function, we measured SADBE-induced ear edema in

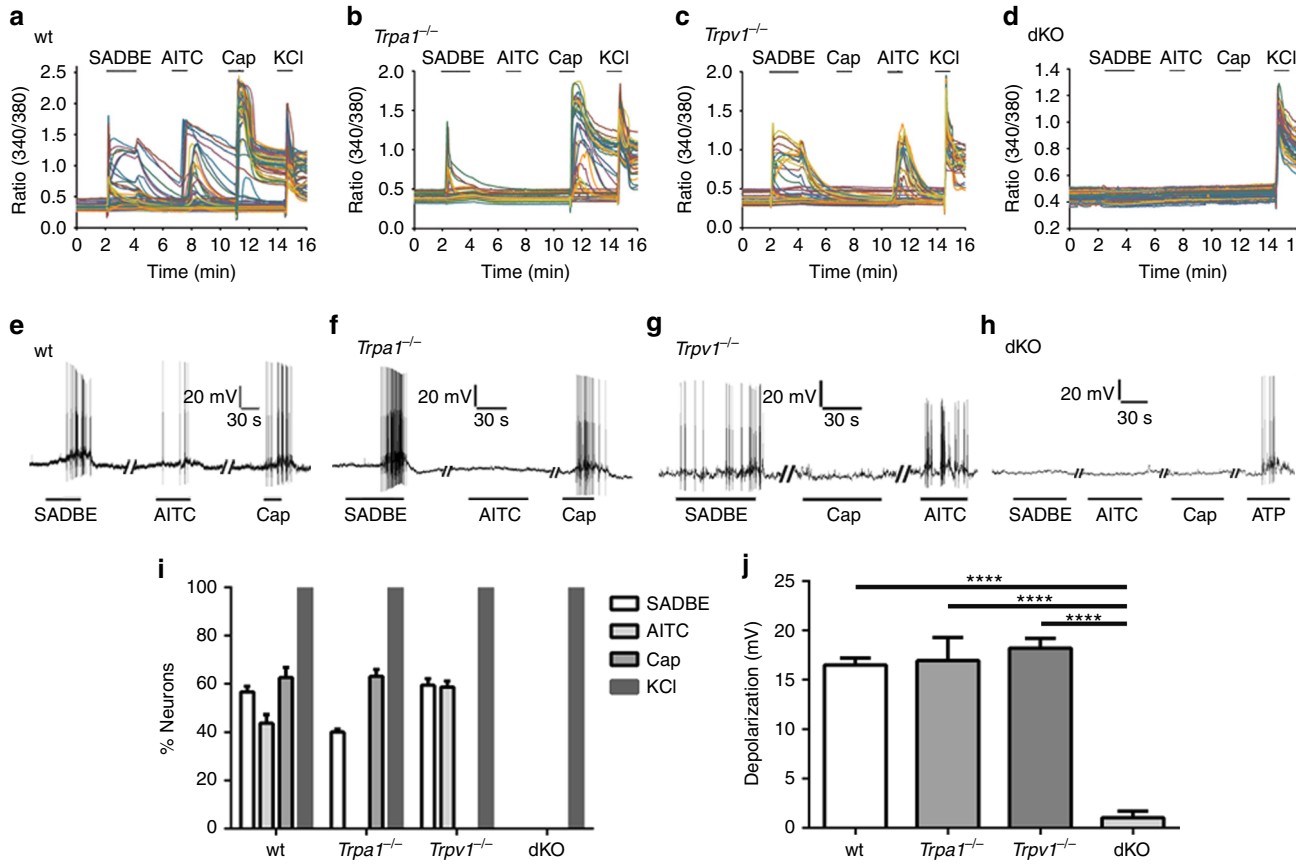

**Fig. 3** Genetic ablation of TRPA1/TRPV1 abolishes SADBE-induced activation of DRG. **a–d** Representative time-lapse traces show that SADBE-induced a robust $[Ca^{2+}]_i$ response in 56.7% of wt (317/558, **a**), 40.2% of $Trpa1^{-/-}$ (220/547, **b**), 59.5% of $Trpv1^{-/-}$ (334/561, **c**), and 0% of $Trpa1^{-/-}/Trpv1^{-/-}$ dKO (0/509, **d**) DRG neurons. $n = 5$ coverslips for each genotype; **e–h** Representative traces show that SADBE-induced depolarization of membrane potential and action potential firing in DRG neurons isolated from wt (**e**), $Trpa1^{-/-}$ (**f**), $Trpv1^{-/-}$ (**g**) and $Trpa1^{-/-}/Trpv1^{-/-}$ dKO (**h**) mice, $n = 5$ for each genotype; (**i**) Percentages of DRG neurons responding to SADBE, AITC, capsaicin and KCl in neurons isolated from wt, $Trpa1^{-/-}$, $Trpv1^{-/-}$ and $Trpa1^{-/-}/Trpv1^{-/-}$ dKO mice in live-cell $Ca^{2+}$ imaging assays. Data are presented as mean ± SEM; (**j**) Quantification of SADBE-induced depolarization of membrane potentials of DRG neurons isolated from wt, $Trpa1^{-/-}$, $Trpv1^{-/-}$ and $Trpa1^{-/-}/Trpv1^{-/-}$ dKO mice. Data are presented as mean ± SEM. Asterisks indicate statistical significance. ****$p < 0.0001$, ANOVA

the $Trpa1^{-/-}$, $Trpv1^{-/-}$ and $Trpa1^{-/-}/Trpv1^{-/-}$ dKO mice. We found that SADBE-induced ear edema was significantly increased in the $Trpv1^{-/-}$ and $Trpa1^{-/-}/Trpv1^{-/-}$ dKO mice but not the $Trpa1^{-/-}$ mice compared with that in the wt mice (Figs. 6b and d). Consistent with the result from genetic ablation approach, pharmacological inhibition of TRPV1 function using a specific TRPV1 antagonist AMG517 also aggravated the SADBE-induced skin inflammation (Supplementary Fig. 2b). On the other hand, SADBE-induced skin inflammation was not affected by administration of a selective TRPA1 channel blocker A 967079[36], further supporting that TRPV1 but not TRPA1 plays a critical role in modulating ear edema in the SADBE-induced ACD model.

Inflammation is a complex process involving various types of immune cells and signaling molecules. Both Th1 and Th2 cells are shown to be important effector cells playing important r oles in the initiation and maintenance of ACD. It is possible that the aggravation of ear edema in the TRPV1 KO mice might result from enhanced infiltration of Th1 and/or Th2 cells in SADBE-induced ACD in the absence of TRPV1 modulation. To explore this possibility, we analyzed the changes of the Th1 and Th2 populations in the ear preparations dissociated from the wt, $Trpa1^{-/-}$, $Trpv1^{-/-}$ and $Trpa1^{-/-}/Trpv1^{-/-}$ dKO mice following SADBE challenges by flow cytometry. Indeed, the numbers of both CD4$^+$ and CD4$^-$ T cells were increased in both $Trpv1^{-/-}$ and $Trpa1^{-/-}/Trpv1^{-/-}$ dKO mice after SADBE treatments (Supplementary Figs. 7, 8a, 8b and 8d), which is consistent with cutaneous inflammation promoted by applications of SADBE. However, the percentages of neither CD4$^+$ and CD4$^-$ cells in the CD45$^+$ T cell population nor Th1 and Th2 cells in the CD4$^+$ T cell population were significantly increased in the $Trpv1^{-/-}$ mice when compared with that in the wt mice (Supplementary Figs. 7, 8a, 8c, 8e, 8g and 8i), suggesting that T cells were not responsible for the aggravated inflammation in the $Trpv1^{-/-}$ mice.

Besides lymphocytes, myeloid cells also exhibit extensive plasticity in response to inflammatory stimuli. Thus, we next asked if myeloid cells are involved in the aggravated SADBE-induced skin inflammation in the $Trpv1^{-/-}$ mice. Strikingly, we found that both the cell number and the percentage of macrophages but not other myeloid populations including dermal dendritic cells, neutrophils, and NK cells, were significantly increased in both $Trpv1^{-/-}$ and $Trpa1^{-/-}/Trpv1^{-/-}$ dKO mice when compared with that in wt control and $Trpa1^{-/-}$ mice (Figs. 7a, b, c, Supplementary Figs. 7 and 9). These results suggest that macrophages might play a predominant role in the exacerbation of SADBE-induced skin inflammation caused by the absence of TRPV1 modulation.

Macrophages secrete many inflammatory cytokines upon exposure to inflammatory stimuli. All of these molecules, together, may increase vascular permeability and recruit many types of inflammatory cells to the sites of inflammation. Indeed, deregulated cytokine secretion from macrophages is implicated in immune-mediated skin diseases including ACD[37, 38]. Thus, we measured expression levels of inflammatory cytokines in the ear preparations treated with SADBE in the wt, $Trpa1^{-/-}$, $Trpv1^{-/-}$ and $Trpa1^{-/-}/Trpv1^{-/-}$ dKO mice using Real-Time qRT-PCR (Quantitative Reverse Transcription PCR). Consistent with the flow cytometry results, the expression levels of TNFα, IL-1β, and IL-6, which are abundantly expressed by dermal macrophages[37], were significantly higher in both $Trpv1^{-/-}$ and $Trpa1^{-/-}/Trpv1^{-/-}$ dKO but not in the wt and $Trpa1^{-/-}$ ear preparations treated with SADBE (Fig. 7d–f). In marked contrast, the expression levels of Th1 cytokine IFN-γ and Th2 cytokines IL-4, IL-5, and IL-31 were not significantly different among wt, $Trpa1^{-/-}$, $Trpv1^{-/-}$ or $Trpa1^{-/-}/Trpv1^{-/-}$ dKO ear preparations (Supplementary Fig. 10). Together, these data suggest that TRPV1 deficiency, selectively promotes the expression of TNFα, IL-1β, and IL-6 in dermal macrophages in the SADBE-induced ACD model, leading to an exaggerated inflammatory response.

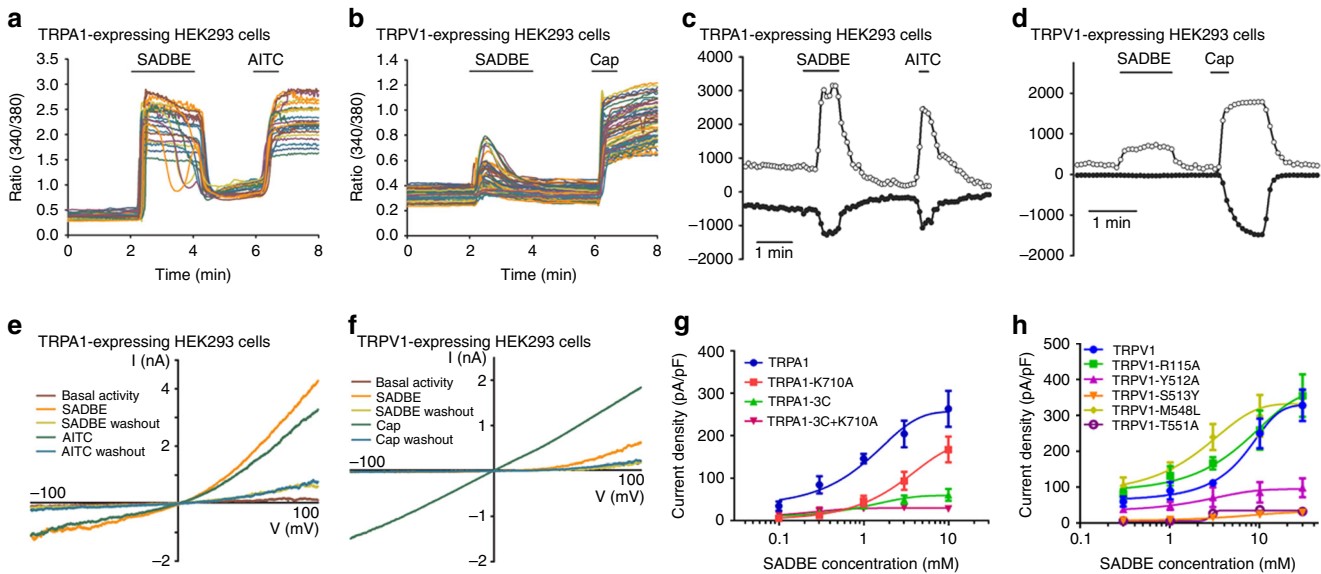

**Fig. 4** Structural basis of SADBE activation of recombinant TRPA1 and TRPV1. **a**, **b** Representative time-lapse traces illustrate SADBE-elicited [Ca$^{2+}$]$_i$ responses in HEK293 cells transfected with TRPA1 (**a**) and TRPV1 (**b**) constructs, which could be also activated by AITC and capsaicin, respectively; **c**, **d** Representative current traces illustrate that SADBE activated large outward (at +100 mV) and inward currents (at −100 mV) in HEK293 cells transfected with TRPA1 (**c**) and TRPV1 (**d**) constructs, which were also activated by AITC or capsaicin, respectively; **e**, **f** Representative I-V curves illustrate that SADBE-activated outwardly rectifying TRPA1 (**e**) and TRPV1 (**f**) currents. Note that the TRPA1-expressing HEK293 cell was activated by AITC and the TRPV1-expressing cell was activated by capsaicin; **g** Concentration–response relationships of SADBE in wild-type and mutant TRPA1 channels; **h** Concentration–response relationships of SADBE in wild-type and mutant TRPV1 channels

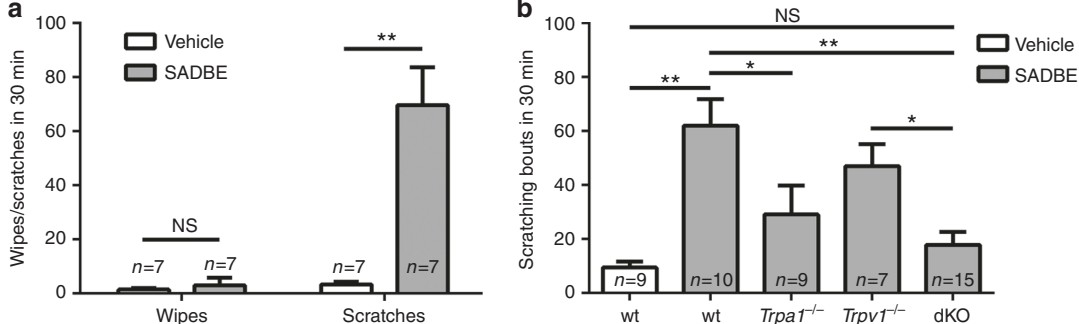

**Fig. 5** TRPA1 is the primary mediator of SADBE-induced acute itch. **a** Bar charts illustrate acute itch and pain behaviors in wt mice by cheek injections of SADBE (30 mM). Facial wiping with the forelimb was indicative of a pain response, whereas scratching with the hind paw was considered as an itch response. *Asterisks* indicate statistical significance. Data are presented as mean ± SEM. NS, not significant. **$p < 0.01$, Student's *t*-test; **b** Bar charts illustrate acute itch responses elicited by intradermal injections of vehicle or SADBE (30 mM) in the neck of wt, *Trpa1*[−/−], *Trpv1*[−/−] and *Trpa1*[−/−]/*Trpv1*[−/−] dKO mice. Data are presented as mean ± SEM. NS, not significant. *Asterisks* indicate statistical significance. *$p < 0.05$; **$p < 0.01$, ANOVA

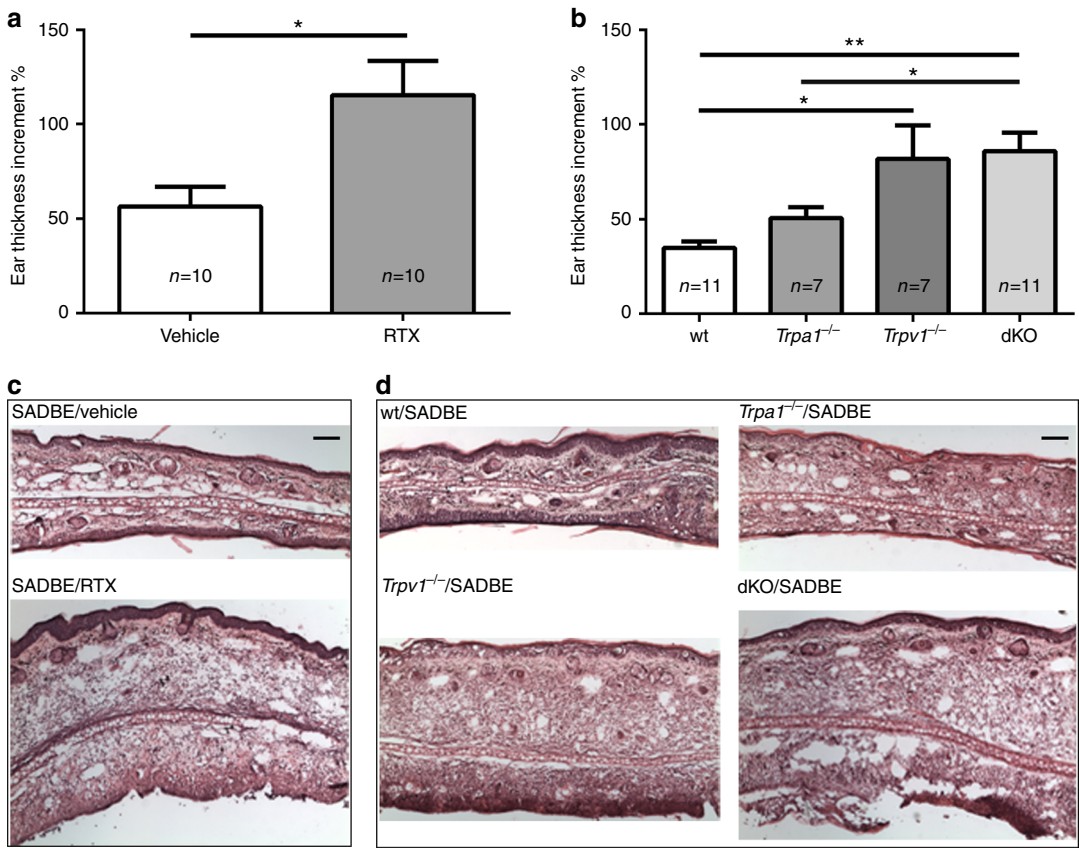

**Fig. 6** TRPV1-expressing sensory nerves protect against skin inflammation. **a** Bar charts illustrate ear thickness increment in wt mice pre-treated with either vehicle or RTX. Data are presented as mean ± SEM. *Asterisks* indicate statistical significance. *$p < 0.05$, Student's *t*-test; **b** Bar charts illustrate ear thickness increment in wt, *Trpa1*[−/−], *Trpv1*[−/−] and *Trpa1*[−/−]/*Trpv1*[−/−] dKO mice. Data are presented as mean ± SEM. *Asterisks* indicate statistical significance. *$p < 0.05$; **$p < 0.01$, ANOVA; **c** Representative H&E staining of ear skin sections from wt mice, pre-treated with vehicle or RTX; *Scale bar*, 100 μm; **d** Representative H&E staining of ear skin sections from wt, *Trpa1*[−/−], *Trpv1*[−/−] and *Trpa1*[−/−]/*Trpv1*[−/−] dKO mice. *Scale bar*, 100 μm

## Discussion

Chronic itch is a common symptom of numerous cutaneous and systemic diseases with unmet medical needs. A better understanding of the molecular and cellular mechanisms underlying chronic itch is critical to the development of effective and safe therapeutic strategies to provide relief for this debilitating skin condition. Our data provided the first evidence that both TRPA1 and TRPV1 channels are critical mediators of persistent itch in the mouse model of SADBE-induced contact dermatitis, which is independent of the lymphocyte-mediated immunity. On the other hand, TRPV1 functions as a potent modulator of skin inflammation induced by pre-sensitization and repeated SADBE challenges (Fig. 8). By using live-cell Ca²⁺ imaging and whole-cell patch clamp recordings we have shown that SADBE directly activates both TRPA1 and TRPV1 channels through interactions with several key amino acid residues known to be essential to

activation of TRPA1 and TRPV1 by pungent chemical irritants. Our findings suggest that TRPA1 and TRPV1 contribute differentially to the pathogenesis of persistent itch and skin inflammation in the SADBE-induced CHS.

Application of SADBE elicited a large intracellular $Ca^{2+}$ response and activated membrane currents in HEK293 cells transfected with either TRPA1 or TRPV1 but not vector control, TRPV3, or TRPV4, suggesting a direct activation of these sensory channels by SADBE. We further showed that point mutations disrupting TRPA1 activation by AITC or TRPV1 activation by vanilloids either abolished or severely attenuated SADBE-activated TRPA1 or TRPV1 responses in heterologous cells,

providing the molecular basis of SADBE activation of these two sensory TRP channels. SADBE evoked a robust intracellular $Ca^{2+}$ response and promoted membrane excitability of freshly isolated wt DRG neurons, as reflected by increased membrane depolarization and action potential firing upon applications of SADBE. Although the intracellular $Ca^{2+}$ response, action potential firing, and membrane depolarization induced by SADBE applications were completely absent in the DRG neurons from the $Trpa1^{-/-}/Trpv1^{-/-}$ dKO mice, they persisted in the DRG neurons isolated from single $Trpa1^{-/-}$ or $Trpv1^{-/-}$ mice, presumably resulting from mutual compensatory actions from these two

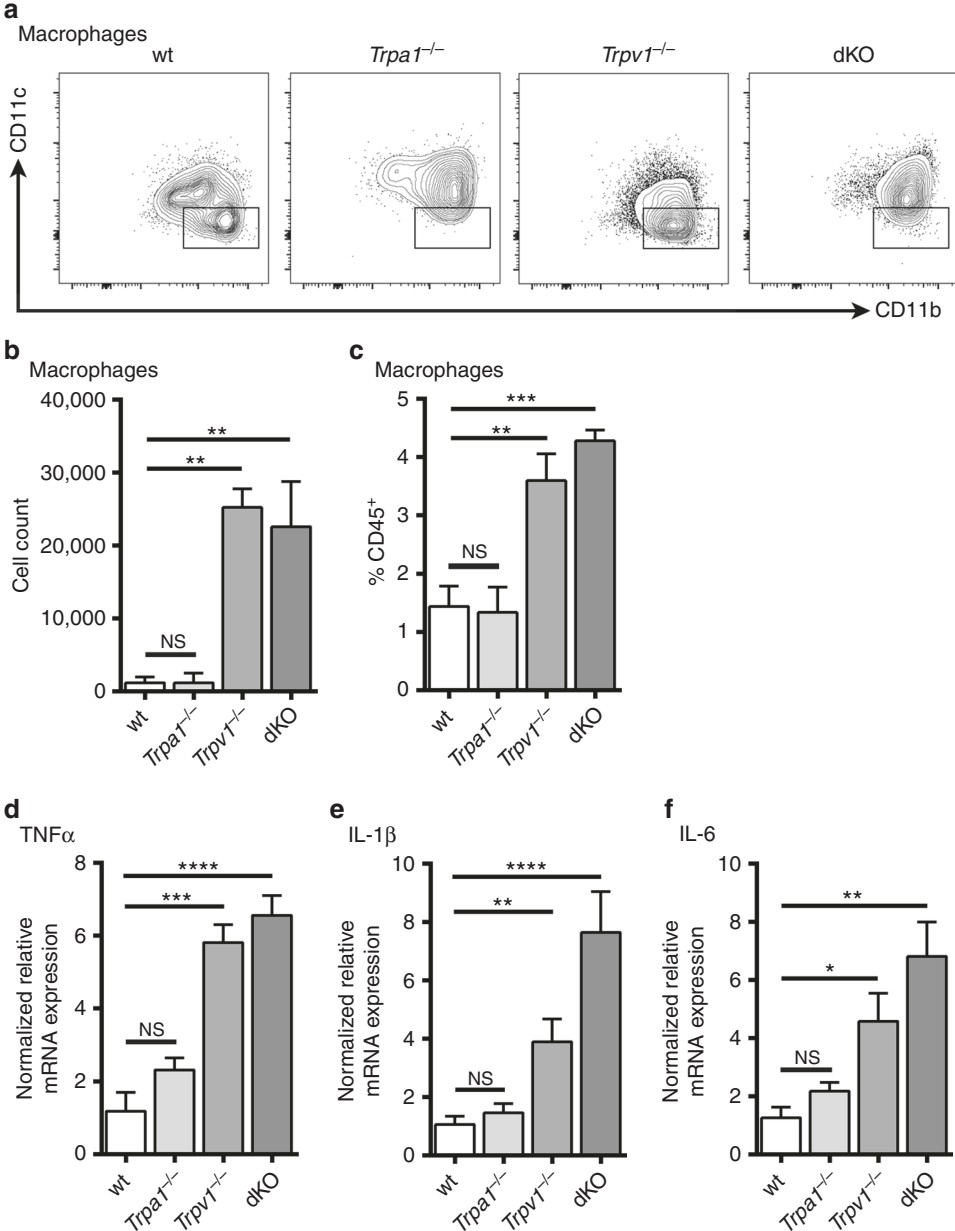

**Fig. 7** TRPV1 deficiency promotes the function of dermal macrophages. **a** Representative FACS plots of macrophages in the ear preparations isolated from wt, $Trpa1^{-/-}$, $Trpv1^{-/-}$, and $Trpa1^{-/-}/Trpv1^{-/-}$ dKO mice. Macrophages were defined as I-A$^{b-lo/-}$ F4/80$^+$ CD11b$^+$ CD11c$^-$; **b**, **c** Quantification of the number (**b**) and the percentage (**c**) of macrophages based on flow cytometry analysis. $n = 3$. Data are presented as mean ± SEM. NS, not significant. *Asterisks* indicate statistical significance. $**p < 0.01$; $***p < 0.001$, ANOVA; **d–f** Expression of proinflammatory cytokines TNFα (**d**), IL-1β (**e**) and IL-6 (**f**), in the SADBE-treated ear preparations of wt, $Trpa1^{-/-}$, $Trpv1^{-/-}$, and $Trpa1^{-/-}/Trpv1^{-/-}$ dKO mice. Data are presented as mean ± SEM. NS, not significant. *Asterisks* indicate statistical significance. $*p < 0.05$; $**p < 0.01$; $***p < 0.001$; $****p < 0.0001$, ANOVA

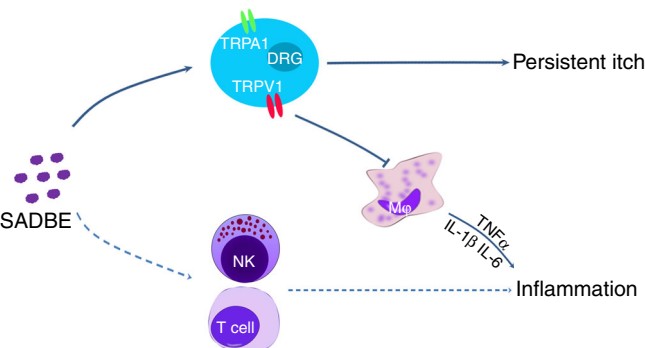

**Fig. 8** Working model for SADBE-induced CHS. SADBE could produce persistent itch by directly activating TRPV1-expressing primary sensory neurons that modulates skin inflammation through regulating the function of dermal macrophages. Dash lines represent the classic lymphocytes mediated-inflammation pathway in ACD

sensory channels. Combined, these results strongly suggest that SADBE directly activates both TRPA1 and TRPV1 channels.

Persistent itch has been considered as an accompanying irritating symptoms of many types of chronic inflammatory skin disorders including atopic dermatitis (AD), ACD, and psoriasis and management of itch is generally assumed to overlap with that for the skin inflammation[29]. Indeed, previous studies have shown that activation of the sensory TRPA1 channels could promote skin edema, epidermal hyperplasia, and scratching behavior through releasing sensory neuropeptides to regulate skin immune responses in mice treated with oxazolone and urushiol, haptens known to cause CHS[17]. Moreover, clinical treatment for skin inflammation could indeed improve chronic itch in a number of itch-related skin disorders including AD[39–41]. On the other hand, chronic itch could be secondary to systemic diseases without skin inflammation or primary skin lesions, such as cholestasis and chronic renal failure[42–44]. In addition, when immunological therapies showed promising results to curb skin inflammation their efficacy to improve chronic itch was clearly lower than that for the inflammation in AD patients[45]. Furthermore, a rapid onset of treatment effect on itch was observed as early as within 1 day of starting treatment whereas the improvement for inflammation generally developed slowly after treatment with Tofacitinib, a janus kinase inhibitor, in psoriasis patients[46]. These observations point to a potential mechanistic difference between itch and inflammation in chronic inflammatory skin disorders. These clinical observations are also consistent with our surprising finding that the spontaneous scratching in mice following SADBE challenges was not affected in mice lacking all lymphocytes that are required for generating the SADBE-induced skin inflammation. Moreover, SADBE could produce inflammation and itch in the affected skin area in mice when directly applied to the ear without prior sensitization. Our results strongly suggest that SADBE produces both allergic and irritant contact dermatitis in mice. While the skin edema results from lymphocyte-mediated immune response the persistent itch likely also involves non-immunological reactions similar to that often seen in the irritant contact dermatitis[20].

Primary sensory nerve fibers are required to elicit a CHS response and denervation of sensory nerve fibers can completely abolish CHS responses induced by some haptens[47]. Previous studies have also elegantly demonstrated that both MrgprA3- and MrgprD-positive neurons displayed enhanced excitability[6] and CXCR3/CXCL10 signaling pathway contributed to allergic itch sensation in the SADBE model[15]. In this study, we have provided strong evidence showing that the sensory TRPA1 and TRPV1

channels are both required to produce persistent itch induced by SADBE challenges. Interestingly, both MrgprA3 and MrgprD are expressed in the TRPV1-positive DRG neurons which also co-express TRPA1[25, 26, 48, 49]. Although only a small population (about 20%) of the MrgprD-expression neurons express TRPV1 and MrgprD is a mediator for both pain and itch sensations, most of the MrgprA3-expressing DRG neurons are TRPV1-positive and MrgprA3 is a specific itch receptor[48–51]. We speculate that the TRPA1 and TRPV1 are molecular itch sensors in the MrgprA3- and potentially also in the MrgprD-positive DRG neurons and promotes membrane excitability and action potential firing in these itch-mediating neurons following SADBE challenges.

Interestingly, although both TRPA1 and TRPV1 mediate persistent itch in the SADBE-induced CHS, TRPA1 but not TRPV1 mediates the acute scratching behavior elicited by intradermal injection of SADBE. One explanation for the discrepancy of TRPV1-mediated responses in the acute and chronic settings is that SADBE might be too weak an activator of TRPV1 to elicit a TRPV1-dependent acute itch when injected intradermally into the healthy wt mice. On the other hand, TRPV1 function is known to be sensitized under inflammatory states such as in SADBE-induced ACD, which could decrease the threshold of SADBE-induced activation of TRPV1 and produce TRPV1-dependent spontaneous scratching behavior.

Although TRPA1-deficiency was shown to reduce skin inflammation in the oxazolone-induced CHS[17], the SADBE-induced cutaneous inflammatory response persisted in the TRPA1 KO mice, suggesting that TRPA1 is differentially involved in the pathogenesis of skin inflammation elicited by different haptens. Unexpectedly, we found that TRPV1 deficiency could promote the infiltration of macrophages and increase the expression of TNFα, IL-1β, and IL-6, which is consistent with the observations that TRPV1 deficiency or pharmacological ablation of the TRPV1-positive sensory fibers promoted skin inflammation in the SADBE-induced CHS. These findings are also consistent with previous studies showing that the primary sensory neurons are actively engaged in regulating the functions of many skin resident cells such as dermal dendritic cells and keratinocytes through releasing bioactive substances including neuropeptides to produce neurogenic inflammation and regulate cutaneous immunity[18, 28, 52].

During the sensitization and challenge phases, there is an extremely complex interaction among different types of immune cells, such as antigen-presenting cells including macrophages, antigen-specific T lymphocytes, NK cells, and peripheral blood leukocytes (monocytes and neutrophils). While the importance of antigen-specific T cells for the induction of ACD is apparent, contributions of other immune cells may differ in a hapten-specific manner. In this study, we found that dermal macrophages play a critical role in the exacerbation of the inflammation in the TRPV1-deficient mice as the number of dermal macrophages and the expression of macrophage-derived cytokines including TNFα, IL-1β, and IL-6 are selectively increased in these animals but not in wt controls or the TRPA1-deficient mice. These results were consistent with previous studies that macrophages accumulate and produce high levels of proinflammatory cytokines at the sites of hapten-induced ACD[37]. It was reported that the induction of neuropeptides, such as calcitonin gene-related peptide (CGRP) and substance P (SP), is tightly correlated with the activity of sensory neurons[53, 54]. Although it is still unclear how TRPV1 channels control the function of macrophages in SADBE-induced inflammation, we speculate that CGRP but not SP is the most likely candidate mediating the SADBE action as previous studies have shown that CGRP strongly inhibits macrophage function while SP enhances macrophage activity[55, 56].

In summary, our studies present evidence that the SADBE-induced persistent itch involves distinct signaling pathways separable from that mediating skin inflammation. The protective role of TRPV1 through modulating the immune function of macrophages in SADBE-induced inflammation has further highlighted functional heterogeneity of interactions between the primary sensory nerves and the cutaneous immune system (Fig. 8). Recognizing the distinct roles of TRP channels in the pathogenesis of skin inflammation and chronic itch will help to develop effective treatment strategies for chronic itch in ACD by targeting specific TRP channels. Moreover, extra caution should be taken to avoid unwanted deleterious side effects when using TRP channel blockers to treat CHS in ACD because the same TRP channel, for instance, TRPV1, might play distinct roles in skin inflammation and itch.

## Methods

**Animals.** C57BL/6 J and Rag1[−/−] mice were obtained from Jackson Laboratories (Bar Harbor, ME, USA). Trpv1[+/+] and congenic Trpv1[−/−] mice on the C57BL/6 J background were obtained from Jackson Laboratory (Bar Harbor, ME, USA). Trpa1[+/+] and congenic Trpa1[−/−] mice on the C57BL/6 J background were described previously[57]. Both Trpv1[−/−] and Trpa1[−/−] mice were continuously backcrossed to C57BL/6 J. Trpa1[−/−]/Trpv1[−/−] dKO mice were generated by crossing Trpv1[−/−] and Trpa1[−/−] animals. All mice were housed under a 12 h light/dark cycle with food and water provided ad libitum. All behavioral tests were videotaped from a side angle, and behavioral assessments were done by observers blind to the treatments or genotypes of animals. All mice used for behavior tests were genotyped and allocated to experimental groups or control groups. Body weight- and gender-matched wt, Trpa1[−/−], Trpv1[−/−] and Trpa1[−/−]/Trpv1[−/−] dKO mice were used at 12-weeks old for all the experiments. Rag1[−/−] mice were used at 10-weeks old. All experiments were performed in accordance with the guidelines of the National Institutes of Health and the International Association for the Study of Pain, and were approved by the Animal Studies Committee at Washington University School of Medicine.

**HEK293 cell culture and transfection.** HEK293 cells were purchased from ATCC (ATCC CRL-1573). Cells were tested for mycoplasma contamination before culturing in lab and grown as a monolayer maintained in DMEM (Life Technologies, Carlsbad, CA, USA), supplemented with 10% FBS (Life Technologies), 100 units/ml penicillin, and 100 μg/ml streptomycin in a humidified incubator at 37 °C with 5% $CO_2$. cDNAs for mouse TRPV1 (mTRPV1), individual mTRPV1 mutants, human TRPA1 (hTRPA1), individual hTRPA1 mutants, mouse TRPV3 (mTRPV3), or rat TRPV4 (rTRPV4) were transiently transfected to HEK293 cells for at least 24 h using Lipofectamine 2000 (Invitrogen, Carlsbad, CA, USA). QuikChange II XL Mutagenesis Kit (Agilent Technologies, Inc., Santa Clara, CA, USA) and DNA sequencing were performed to make all the mutants.

**Overnight culture of mouse DRG neurons.** Murine DRG were isolated and cultured using a previous described protocol[58]. In brief, laminectomies were performed on mice and bilateral DRG were dissected out. After removal of connective tissues, DRG were digested with 1 mL $Ca^{2+}/Mg^{2+}$-free Hank's Balanced Salt Solution (HBSS) containing 1 μL saturated $NaHCO_3$, 0.35 mg L-cysteine, and 20 U papain (Worthington Biochemical, Lakewood, NJ, USA) for 10 min in 37 °C water bath. After centrifugation, the supernatants were further treated with 1 mL $Ca^{2+}/Mg^{2+}$-free HBSS containing 3.75 mg collagenase type II (Worthington Biochemical, Lakewood, NJ, USA) and 7.5 mg dispase (Worthington Biochemical) for 15 min. Neurons were gently triturated, pelleted, and then resuspended in Neurobasal-A culture medium containing 2% B-27 supplement (ThermoFisher Scientific, Waltham, MA, USA), 100 U/mL penicillin plus 100 μg/mL streptomycin (Sigma-Aldrich, St. Louis, MO, USA), 100 ng/mL nerve growth factor (NGF, Sigma), 20 μg/mL glial cell-derived neurotrophic factor (GDNF, Sigma-Aldrich) and 10% heat-inactivated FBS (Sigma-Aldrich). After plating, DRG neurons were kept in a humidified incubator at 37 °C for at least 24 h.

**Live-cell $Ca^{2+}$ imaging.** 4 μM Fura-2 AM (Life Technologies) was used to loading cultured DRG neurons and TRP channel-expressing HEK293 cells were loaded for 60 min. Before use, Fura-2-loaded cells were washed for at least 3 times with HBSS. Fluorescence was recorded at 340 nm and 380 nm excitation wavelengths using an inverted Nikon Ti-E microscope with NIS-Elements imaging software (Nikon Instruments, Inc., Melville, NY, USA). Fura-2 ratios (F340/F380) reflecting changes in $[Ca^{2+}]_i$ upon stimulation were monitored and recorded. Cells were considered responsive if they demonstrated a change in fluorescence ratio >10% of baseline[58].

**Electrophysiology.** Whole-cell patch clamp recordings were performed at room temperature (22–24 °C) using an Axon 700B amplifier (Molecular Devices,

Sunnyvale, CA, USA). For transfected cells, GFP-expressing cells were selected by a microscope equipped with a filter set for GFP visualization (Nikon Instruments Inc., Melville, NY, USA)[58]. For DRG neurons, small diameter neurons were preferred in recording. Pipettes prepared for whole-cell patch clamp recordings were pulled from borosilicate glass (BF 150-86-10; Sutter Instrument, Novato, CA, USA) with a Sutter P-1000 pipette. The intracellular solution contains 140 mM CsCl, 2 mM EGTA, and 10 mM HEPES with pH 7.3. Osmolarity was adjusted to 315 mOsm/l with sucrose. To prevent $Ca^{2+}$-dependent desensitization of TRPV1 currents, a $Ca^{2+}$-free extracellular solution was made with 140 mM NaCl, 5 mM KCl, 0.5 mM EGTA, 1 mM $MgCl_2$, 10 mM glucose, and 10 mM HEPES with pH 7.4 and 340 mOsm/l osmolarity. Holding at 0 mV, voltage ramp from −100 to +100 mV for 500 ms was used to record whole-cell current. Resting membrane potential was recorded with a gap free protocol for each neuron under the current clamp mode after stabilization (within 3 min). A neuron was included only if the resting potential was more negative than −40 mV. For current clamp recordings, the internal solution contained: 140 mM KCl, 1 mM $MgCl_2$, 3 mM MgATP, 1 mM EGTA, 10 mM HEPES, 10 mM Sucrose with pH 7.2 and 290–300 mOsm/l osmolarity. The external solution contained: 140 mM NaCl, 5 mM KCl, 2 mM $CaCl_2$, 1 mM $MgCl_2$, 10 mM HEPES, 10 mM Glucose and 20 mM Sucrose, pH adjusted at 7.4 with NaOH. Data were acquired using Clampex 10.4 software (Molecular Devices). Currents were filtered at 2 kHz and digitized at 10 kHz. Data were analyzed and plotted using Clampfit 10 (Molecular Devices). The concentration–response curve was fitted with the logistic equation: $Y = Ymin + (Ymax−Ymin)/(1 + 10^{[(logEC_{50}−X) × Hill slope]})$, where Y is the response at a given concentration, Ymax and Ymin are the maximum and minimum responses, X is the logarithmic value of the concentration and Hill slope is the slope factor of the curve. $EC_{50}$ is the concentration that gives a response halfway between Ymax and Ymin. All data are presented as mean ± SEM.

**Quantitative RT–PCR.** Total RNA was extracted from mouse ear tissue using RNeasy kit (Qiagen, Germantown, MD, USA) according to manufacturer's instructions. RNA was treated with DNase I (Invitrogen) and the cDNA was synthesized in vitro using ThermoScript RT–PCR System kit (Invitrogen). Reactions were carried out in a volume of 20 μl per reaction containing 10 μl SYBR Green master mix (2×) (Roche, Risch-Rotkreuz, Switzerland), 0.5 μl cDNA, 1.2 μl 5 μM primer mix, and 8.3 μl water using StepOnePlus real-time PCR system (Applied Biosystems, Foster City, CA, USA). Relative mRNA expression levels of different target gene compared to GAPDH were calculated using 2-ΔΔCt methods. Primer sequences used for each gene were selected from pre-validated PrimeTime qPCR Assays (Integrated DNA Technologies, Coralville, Iowa, USA). Primer sequences (5′ to 3′) used were:

TNFα: CCCTCACACTCAGATCATCTTCT (forward) and CTCCTCCACTTGGTGGTTTG (reverse);
IL-1β: TGTAATGAAAGACGGCACACC (forward) and TCTTCTTTGGGTATTGCTTGG (reverse);
IL-6: TCTAATTCATATCTTCAACCAAGAGG (forward) and TGGTCCTTAGCCACTCCTTC (reverse);
IFN-γ: GATATCTGGAGGAACTGGCAAAA (forward) and CTTCAAAGAGTCTGAGGTAGAAAGAGATAAT (reverse);
IL-4: AGATGGATGTGCCAAACGTCCTCA (forward) and AATATGCGAAGCACCTTGGAAGCC (reverse);
IL-5: CTCCAATGCATAGCTGGTGAT (forward) and GAGATTCCCATGAGCACAGT (reverse);
IL-31: TCCTATACAGCAAAGCAGCAC (forward) and CCAGAGACCACAGGCAAAG (reverse).

**Hematoxylin and eosin staining.** Mouse ears were collected and fixed in 4% Paraformaldehyde in phosphate-buffered saline (PBS), dehydrated in 30% sucrose, and embedded in OCT. Twelve-micron sections were generated using a Leica CM1950 cryostat (Leica Biosystems, Buffalo Grove, IL, USA). Hematoxylin and Eosin (H&E) staining was performed according to standard protocols.

**FTY720 treatment and depletion of NK cells.** FTY720 (1 mg/kg), an agonist which results in S1P receptor internalization, or PBS was injected daily intraperitoneally starting from day 0 of SADBE sensitization[28]. Depletion of NK cells was effectuated by weekly i.p. injection of 25 μg anti-NK1.1 antibody (eBioscience, San Diego, CA, USA), starting on the day before SADBE sensitization. Control mice were injected with the appropriate isotype control IgG antibody (eBioscience)[23].

**Mouse model of ACD.** Squaric acid dibutylester (SADBE) (Tokyo Chemical Co., Tokyo, Japan) was used to elicit contact hypersensitivity in the mouse as a model of allergic contact dermatitis in humans. Mice were sensitized by the topical application of 20 μl of 0.5% SADBE in acetone to the shaved abdominal skin once a day for three consecutive days. 5 days later, the SADBE-treated group was challenged with a topical application of 20 μl of 0.5% SADBE to the right ear once a day for three consecutive days whereas acetone alone was used as the vehicle control. 3 days later, the scratching behavior with the hind paw was quantified by recording the number of incidences of scratching bouts for 60 min.

**Acute itch behavior**. For the intradermal injections of SADBE, a 2.3 M stock solution was made by dissolving SADBE in DMSO with a ratio of 1:1 (volume). The final 30 mM SADBE solution was prepared by 1:77 dilutions in 0.9% saline. Mice were shaved on the nape of the neck or in the face 2 days before assay. On the day of experiment, mice were acclimated for 1 h by placing each of them individually in the recording chamber followed by intradermal injection of SADBE to the nape of the neck and/or cheek. Immediately after the injection, mice were video-taped for 30 min without any person in the recording room. After the recording, the videotapes were played back and the number of scratching bouts towards the injection site was counted by an investigator blinded to the treatment.

**Nocifensive behavior**. Intraplantar injection of SADBE was used to induce nociceptive responses. Mice were acclimated for 1 h before recording. Immediately after injection, the licking/lifting behavior was recorded for 10 min and counted by an investigator blinded to the treatment.

**Pharmacological ablation of TRPV1-positive sensory nerves**. Resiniferatoxin (RTX) (Sigma-Aldrich), a capsaicin analog, was injected subcutaneously into the flank of 4 week-old mice in three escalating (30, 70, and 100 µg/kg) doses on consecutive days. Control mice were treated with vehicle solution (DMSO in PBS). Mice were allowed to rest for 6 weeks before behavioral test.

**Measurement of ear edema**. Ear thickness was measured with an engineer's micrometer (Moore and Wright, Sheffield, England) with 0.1 mm accuracy, before ear challenge with SADBE and 3 days after the last challenge. Data were expressed as % increase of ear thickness compared to the initial pre-challenge values.

**Flow cytometry analysis**. Single cell suspensions of ear skin were prepared by incubating dermal side down in 500 uL 0.25 mg/mL Liberase TL for 90 min at 37 °C. After filtration through a 100 µm cell strainer, cell suspensions were stained with ZombieRed viability dye (423109, Biolegend, San Diego, CA, USA) per manufacturer's instructions and then with conjugated antibodies on ice for 30 min. Stained cells were then washed and fixed with Cytoperm/Cytofix solution (BD Biosciences, San Jose, CA, USA) and acquired on an LSR Fortessa X-20 SORP. Data were analyzed using FlowJo software. anti-mouse CD16/32 (2.4G2) was from BioXCell (West Lebanon, NH, USA) (CUS-HB-197) and used at 1:300 dilution. anti-T-bet (4B10) (644803), anti-GATA3 (16E10A23) (653813), anti-mouse CD4 (GK1.5) (100449), anti-mouse CD3ε (145-2C11) (100327), anti-mouse CD45 (30-F11) (103105, 103111), anti-mouse NKp46 (29A1.4) (137617), anti-mouse/human CD11b (M1/70) (101245), anti-mouse CD11c (N418) (117328), anti-mouse Ly-6G (1A8) (127639), anti-mouse F4/80 (BM8) (123146), anti-mouse I-A$^b$ (AF6-120.1) (116419) were from Biolegend and used at 1:300 dilution.

**Statistics**. All data are presented as mean ± SEM for n independent observations. Student's t-test was used to analyze statistical significance between two groups. ANOVA and repeated measures tests were used to test hypotheses about effects in multiple groups occurring over time. $P < 0.05$ was considered significantly different.

**Data availability**. The data that support the findings of this study are available within the article and its supplementary information files and from the corresponding author upon reasonable request.

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

## Acknowledgements
This work was supported partly by grants from the National Institutes of Health, R01GM101218, R01DK103901 (to H.H.), and R01AR070116 (to B.S.K.), Washington University School of Medicine Digestive Disease Research Core Center (NIDDK P30 DK052574), The Center for the Study of Itch of Department of Anesthesiology at Washington University School of Medicine (to H.H.), and National Natural Science Foundation of China grant 81373379 (to S.Y.).

## Author contributions
H.H. conceived and supervised the study; J.F. and H.H. designed the research; J.F. performed the behavior tests and calcium imaging; P.Y. performed the H&E staining and Quantitative RT-PCR; J.F. and D.D. performed the patch clamp recording; M.R.M. performed the flow cytometry analysis; J.F., P.Y., and H.H. analyzed the data; J.F. and H. H. wrote the paper. All authors discussed and revised the manuscript.

## Additional information

**Competing interests:** The authors declare no competing financial interests

