## [Peer Review file · Nature Communications]

Reviewers' comments:

Reviewer #1 (Remarks to the Author):

The manuscript by Feng and co-workers investigates the mechanisms underlying squaric acid dibutylester (SADBE) induced itch and inflammation. Previous studies have shown that SADBE elicits inflammation and itch-like behaviors in mice (Qu et al 2014; Qu et al 2015). The novel feature of the current manuscript is that the authors show that SADBE directly activates both TRPV1 and TRPA1 ion channels that are expressed by sensory neurons. The evidence for this mode of action is good, although there are clearly other important actions of SADBE that are not explored. The finding that an electrophilic compound like SADBE activates TRPA1 is not surprising; it would have been surprising if it didn't. The data suggesting an action via binding to the putative capsaicin binding site in TRPV1 adds a more interesting observation. The current study also points to a role of sensory nerves and TRPV1 in regulating SADBE-evoked inflammation. This is perhaps the most interesting observation in the study but it is not investigated in depth.

I think that there are aspects of the study that need clarification either by provision of further data or by stronger arguments.

1. SADBE activates both TRPV1 and TRPA1, which makes it highly unlikely that the behavioral effect of acute exposure is to induce only itch. Such stimulation will activate all nociceptive neurons and evoke pain, which is consistent with the results of Qu et al. 2014, 2015, which showed as much or more pain-like responses than itch-like responses after SADBE administration. Were pain-like responses measured and what happened in the *Trpa1*^{-/-}, *Trpv1*^{-/-} and *Trpa1*^{-/-}/*Trpv1*^{-/-} mice?
2. The conclusions about the relative roles of TRPA1 and TRPV1 would be strengthened by the addition of pharmacological evidence that selective antagonists recapitulated the effects seen in the knockout mice. Deleting these channels (*Trpa1*^{-/-}, *Trpv1*^{-/-} and *Trpa1*^{-/-}/*Trpv1*^{-/-} mice) may have unexpected 'compensatory' or 'modifying' effects on physiological processes.
3. Scratching was not totally eliminated in the *Trpa1*^{-/-}/*Trpv1*^{-/-} mice (Lines 211 – 218). There was still a substantial response. What is the mechanistic basis for this residual pruritic effect – is it related to other actions on sensory neurons, perhaps increased action potential firing due to enhanced voltage gated sodium currents (Qu et al 2014;) or reduced potassium currents?
4. The mechanism by which the absence of either TRPV1 expressing sensory neurons (RTX-treatment) or functional TRPV1 channels (*Trpv1*^{-/-} mice) increases SADBE-evoked inflammation is not investigated in depth. Other studies have shown opposite effects of RTX treatment on inflammation elicited by some haptens (e.g. Imiquimod – Riol-Blanco et al 2014 Nature 510: 157) and similar effects of TRPV1 inhibition for other haptens (e.g. Oxazolone – Bánvölgyi A et al 2005 J Neuroimmunol 169: 86). There is a similar dichotomy between the conclusion that TRPA1 does not influence inflammation in this SADBE model but is important for oxazolone-evoked inflammation (Liu et al., 2013 FASEB J 27:3549). The authors provide some evidence for cytokine changes to explain their findings but the mechanisms by which sensory neurons drive these changes and why this is linked to TRPV1 and not to TRPA1 are not addressed. I recommend that the authors add more details or amend wording to aid the reader.
5. Methods. Lines 439 – 453, 459 - 462 and e.g. Figure 1. A). The protocols for administration of SADBE need to be consistently explained in the methods and the figures. The diagrams in the figures show SADBE administered first to the abdomen and then to the ear. But the scratching behavior is measured after SADBE is administered to either the nape of the neck. B) The timing of SADBE administration when measuring ear edema is unclear. The method (line 461) states measurements made after the challenge at day 3, but the protocol shown in figure 1 shows measurement 3 days after the last SADBE treatment. Do you mean 'measurements were made 3 days after the last SADBE challenge?'
6. What vehicle controls were used in the in vitro experiments and in the intradermal injection experiments? Was it acetone? If so, what was the final acetone concentration and did this evoke scratching. Vehicle controls should be presented.

7. Line 399 onwards - Methods. There are no methods given for the current clamp electrophysiology experiments. These should be added.
8. Line 111. Was there any measurement to show that the FTY720 treatment was effective in blocking T cell migration?
9. Lines 155 – 164. The results shown in Figure 3, panels a-d need to be expressed quantitatively. How many neurons were studied, how many neurons responded to SADBE, how many coverslips? What were the percentages - the term 'similar percentages' is vague? The 3mM SADBA concentration used in these DRG experiments is a sub-maximally effective concentration for TRPV1 activation in the heterologous expression experiments (see Figure 4g). From Fig 3b it looks as though there are many TRPV1 expressing neurons that did not respond to SADBE. Were there SADBE responsive neurons that were not AITC or capsaicin sensitive in DRG neurons from wild-type, *Trpa1*^{-/-} or *Trpv1*^{-/-} mice? What was the percentage of responding neurons in the double knockout (*Trpa1*^{-/-}/*Trpv1*^{-/-}) DRG neurons – was it really 0%?
10. Figure 3, panels b and c. The time courses of the responses in the *Trpa1*^{-/-} and *Trpv1*^{-/-} mouse DRG neurons looks very different (*Trpa1*^{-/-} transient, *Trpv1*^{-/-} more persistent). Was this a consistent finding? Two components of response time courses also appear to be present in the wild-type DRG neurons. A difference in time course would be consistent with the time courses shown for TRPV1- and TRPA1-mediated calcium responses in Figure 4a and b.
11. Lines 191 – 194 & Figure 4. The SADBE-evoked responses of hTRPA1 mutants are depressed, but is this due to a reduced expression level? For human TRPA1 3C mutants, responses to a non-electrophilic agonist such as carvacrol would demonstrate that the channels were well expressed.
12. Lines 197 – 198 and Figure 4. The concentration response curve for the TRPV1 M548L mutant appears to be shifted leftwards compared to the wild-type TRPV1, yet the authors say there was no effect. Some quantification of results and evidence of reproducibility should be added to justify the conclusion.
13. Lines 209 – 210. Presumably there are some words missing in this sentence (e.g. mice lacking functional TRPA1 and/or TRPV1).

Reviewer #2 (Remarks to the Author):

This study shows distinct contributions of TRP channels to skin inflammation and itch in the SADBE-induced contact dermatitis model. By using TRPA1 knockout (KO), TRPV1 KO and TRPA1/TRPV1 double KO mice, the authors demonstrate that persistent scratching caused by SADBE treatment is mediated through both TRPA1 and TRPV1 channels. Calcium imaging and electrophysiological experiments using acutely dissociated DRG cells and HEK cells transfected with TRPA1 or TRPV1 show that SADBE can activate directly both TRP channels. Unlike persistent scratching induced by repeated application of SADBE, acute scratching induced by intradermal SADBE injection is mediated mainly through TRPA1. In contrast to scratching behavior, deficiency or pharmacological inhibition of TRPV1 exacerbates skin inflammation probably through promoting Th1 cytokines. These findings strongly suggest that SADBE causes itch independent of skin inflammation. Although most experiments were appropriately conducted and the manuscript is well organized, there remain a number of questions that should be addressed.

1. The authors found an increase in scratching at 3 days after the final SADBE challenge in sensitized mice (Fig. 1), thereby regarding this scratching as "persistent" itch. On the other hand, they show that intradermal injection of SADBE elicited "acute" scratching in naïve mice (Fig. 5). I strongly recommend that the authors should present the time course of scratching after SADBE application in both the "persistent" and "acute" model, and also should examine the effects of TRP channel deficiency on scratching at different time points. Perhaps, increased scratching will be observed immediately after the final application of SADBE in sensitized mice; does the deficiency of TRPA1 alone or both TRPA1 and TRPV1 inhibit this scratching response? Also, how long does the

scratching induced by SADBE injection in naïve mice continue? The present results raise an interesting question whether persistent scratching is due to continuation of direct activation of TRPA1 and/or TRPV1 by SADBE existing in the skin, or whether other mediator(s) induced by repeated SADBE application eventually activate TRPA1 and TRPV1 channels.

2. As shown in Figs. 1e and i, SADBE challenge on the nape of the neck causes scratching, irrespective of systemic presensitization on the abdominal skin, although skin inflammation is significantly but very weakly promoted by the presensitization. Therefore, SADBE causing symptoms, especially itch (or scratching), should be considered as irritant contact dermatitis (ICD), but not allergic contact dermatitis (ACD).

3. The authors showed that intradermal SADBE injection in naïve mice exclusively caused scratching compared to wiping behaviors (Fig. 5); however, each behavior should be compared to when the vehicle was injected. Scratching and wiping behaviors should not be compared to each other. Furthermore, Qu et al. have reported that repeated SADBE application induces not only itch-related scratching but also pain-like behaviors, such as wiping and licking (Qu L, Brain, 2014). As described, both TRPA1 and TRPV1 play a critical role in pain. Have you checked pain behaviors in your SADBE model? How does the TRP channel deficiency affect pain caused by SADBE treatment?

4. Regarding statistics: (1) when comparing among wild type, TRPA1 KO, TRPV1 KO and double KO mice, the authors should compare all pairs of groups. Most of the comparisons were done only vs. the wild type group. (2) The significant differences in Fig. 3j seem to be wrong. (3) The authors should perform statistical analysis in studies shown in Figs. 4g and h.

5. How did you decide the concentration and dose of SADBE (in vitro: 3 mM; in vivo: 30 mM)? In addition, the authors should present the volume of SADBE solution injected intradermally and what the vehicle was.

6. Calcium imaging and electrophysiological experiments clearly showed mutual compensatory effect in the DRG neurons isolated from TRPA1 KO or TRPV1 KO mice (Fig. 3). On the other hand, SADBE-induced persistent scratching could be suppressed in either TRPA1 KO or TRPV1 KO mice (Fig. 2). If DRG neurons play a dominant role in SADBE-induced persistent itch as shown in Fig. 8, these results would be theoretically inconsistent. Is it possible that TRPA1- and/or TRPV1-expressed in other cells besides DRG neurons is involved in the persistent itch? The authors should fully address this question.

7. The responses to SADBE in TRPV1-expressing HEK293 cells were clearly weaker than those in TRPA1-expressing cells (Fig. 4). Indeed, the EC₅₀ value for TRPV1-expressing cells was 5.6 times higher than that for TRPA1 (1.30 mM for TRPA1 vs. 7.26 mM for TRPV1). Why is that? Also, what concentration of SADBE did you use? If it was 3 mM similar to other in vitro experiments, the authors should use a higher (or submaximal) concentration (e.g., 10 mM) to determine the SADBE activation of TRPV1.

8. The authors suggest that increased Th1 cytokines would contribute to aggravation of SADBE-induced skin inflammation by TRPV1 deficiency. On the other hand, SADBE-induced skin inflammation was not affected in Rag1^{-/-} mice, which lack T cells (Fig. 1f), suggesting that T cells are hardly required for SADBE-induced skin inflammation. Also, since the authors have not examined the effect of NK cell depletion alone on skin inflammation, the role of NK cells and T cells in skin edema by SADBE remains unclear. They thus should further address this question and revise the working model (Fig. 8) to fit their findings.

Reviewer #3 (Remarks to the Author):

Jing et al. showed that the SADBE-induced persistent itch was not depend on lymphocytes, but mediated by TRPA1 and TRPV1channels. They demonstrated SADBE can directly activate both TRPA1 and TRPV1 in vivo using freshly isolated DRG cells and TRPA/TRPV1-expressing HEK293 cells. Further, their observation suggests that TRPV1 also affect SADBE-induced ear swelling via inhibiting the production of Th1 cytokines. In this paper, data were well-presented; however, I had concerns relating to the interpretation of some of the results, and lack of mechanistic insight. Followings are my specific comments:

1. Fig.1e showed that irritant response (innate response) by SADBE challenge was very strong with this experimental protocol. It makes hard to evaluate adaptive immune response in this condition. Author need to modify the protocol to reduce irritant response to SADBE.
2. Authors need to explain why ear swelling was not attenuated in Rag1-deficient (Fig.1f) and FTY720-treated (Fig.1g) mice compared to WT controls. It seems that authors just failed to induce adaptive immune response in these experiments.
3. Authors need to explain the interpretation of the result shown in Fig.1f. Does this result suggest SADBE-induced CHS response is mediated by NK cells but not by T/B cells? If so, is there an antigen-specificity in this response?
4. In Fig.1e, ear thickness increment looks over than 100% in SADBE-sensitized and SADBE-challenged group; however, the increment in same group looks less than 80% in Fig.1f and Fig.1g. Moreover, it was less than 50% in Fig. 6b. What causes these discrepancies?
5. Authors demonstrated that SADBE can directly activate TRPA1/TRPV1 channels. However, they did not present any data evaluating the indirect effect of SADBE; for instance, keratinocytes, mast cells, ILCs, which can be activated by SADBE might subsequently activate TRP channels, as authors described.
6. TRPA1 and TRPV1 play roles in SADBE-induced scratching behavior and ear swelling via Th1 cytokine production, respectively; although in-vivo data suggest that their function in calcium influx is compensable in response to SADBE. Authors need to discuss the mechanistic insight of this discrepancy.
7. Fig7: How about the expression level of IFN-gamma, the most important Th1 cytokine.
8. Fig8: In this paper, there is no data that demonstrate the involvement of DDC in SADBE-induced CHS. Therefore, DDC function shown in Fig.8 seems overspeculation.

Reviewers' comments:

Reviewer #1 (Remarks to the Author):

The manuscript by Feng and co-workers investigates the mechanisms underlying squaric acid dibutylester (SADBE) induced itch and inflammation. Previous studies have shown that SADBE elicits inflammation and itch-like behaviors in mice (Qu et al 2014; Qu et al 2015). The novel feature of the current manuscript is that the authors show that SADBE directly activates both TRPV1 and TRPA1 ion channels that are expressed by sensory neurons. The evidence for this mode of action is good, although there are clearly other important actions of SADBE that are not explored. The finding that an electrophilic compound like SADBE activates TRPA1 is not surprising; it would have been surprising if it didn't. The data suggesting an action via binding to the putative capsaicin binding site in TRPV1 adds a more interesting observation. The current study also points to a role of sensory nerves and TRPV1 in regulating SADBE-evoked inflammation. This is perhaps the most interesting observation in the study but it is not investigated in depth. I think that there are aspects of the study that need clarification either by provision of further data or by stronger arguments.

1. SADBE activates both TRPV1 and TRPA1, which makes it highly unlikely that the behavioral effect of acute exposure is to induce only itch. Such stimulation will activate all nociceptive neurons and evoke pain, which is consistent with the results of Qu et al. 2014, 2015, which showed as much or more pain-like responses than itch-like responses after SADBE administration. Were pain-like responses measured and what happened in the *Trpa1^{-/-}*, *Trpv1^{-/-}* and *Trpa1^{-/-}/Trpv1^{-/-}* mice?

Response: Thanks for bringing up this important point. According to your suggestions, we have performed additional experiments investigating SADBE-elicited pain-like behavior. Surprisingly, we didn't find evident wiping behavior elicited by injections of 30 mM SADBE using the cheek model as shown in the revised manuscript (Figure 5a). Furthermore, we didn't observe any paw licking or flicking behavior within the first 10 mins after intraplantar injections of 30 mM SADBE (data not shown).

To further exclude the possibility that SADBE contributes to mechanical and thermal pain sensitivities, we have performed Von Frey and Hargreaves tests after paw injections of 30 mM SADBE. As shown in Figures 1a and 1b below, we did not find significant differences in paw withdrawal threshold to mechanical stimuli or paw withdrawal latency to heat stimuli compared with mice injected with vehicle only. These results suggest that SADBE acts primarily as an itch-evoking compound when applied acutely at the given dosage.

We further tested pain-like behavior in the SADBE-induced allergic contact dermatitis model. As shown in Figures 1e and 1f below, no wiping behavior was observed while scratching behavior gradually increased during the first 3 days when compared with the vehicle control

group. However, excessive scratching produced excessive skin lesions at day 4 (Figure 1c and 1d) where scratching behavior was markedly reduced while wiping behavior started to show up, suggesting that tissue damage and injury promote pain response and suppress itch response, which is consistent with previous findings that pain sensation constitutively suppresses itch sensation [1,2].

We used 20 μ l of 0.5% SADBE compared with 25 μ l of 1% SADBE used by Qu et al. in their original papers. Our protocol yielded a much better itch response because the low concentration of SADBE we used has delayed the development of pain behavior caused by excessive scratching and prolonged the chronic phase of itch. Taken together, these results suggest that SADBE does not directly elicit acute pain or chronic pain sensation in *wt* mice at the concentrations used in our studies.

[1] Liu Y, et al. "VGLUT2-dependent glutamate release from nociceptors is required to sense pain and suppress itch." *Neuron* 68.3 (2010): 543-556.

[2] Lagerström MC, et al. "VGLUT2-dependent sensory neurons in the TRPV1 population regulate pain and itch." *Neuron* 68.3 (2010): 529-542.

Figure 1. Measuring pain-like behaviors in mice with intraplantar injections of SADBE and SADBE-induced allergic contact dermatitis. (a-b) Paw withdrawal threshold in response to mechanical stimuli (a) and thermal stimuli (b) after intraplantar injections of 10 μ l vehicle or 30 mM SADBE in *wt* mice. n=5 per group. n.s, not significant, Student's t test; (c-d) Representative images of the cheek areas of *wt* mice challenged with SADBE on day 3 (c) and day 4 (d); (e-f) Time courses of wiping (e) and scratching (f) responses after SADBE challenges. n=6 per group. n.s, not significant, ** $p < 0.01$, *** $p < 0.001$, **** $p < 0.0001$, ANOVA.

2. The conclusions about the relative roles of TRPA1 and TRPV1 would be strengthened by the addition of pharmacological evidence that selective antagonists recapitulated the effects seen in the knockout mice. Deleting these channels (*Trpa1*^{-/-}, *Trpv1*^{-/-} and *Trpa1*^{-/-}/*Trpv1*^{-/-} mice) may have unexpected 'compensatory' or 'modifying' effects on physiological processes.

Response: Thanks for this constructive comment. Per your suggestion, we have performed additional experiments and tested the effect of a selective TRPA1 antagonist A967079 and a selective TRPV1 antagonist AMG517 on SADBE-induced CHS.

As expected, the number of scratching bouts was significantly reduced in mice treated with A967079, AMG517 or a combination of A967079 and AMG517 (A967079/AMG517) (Supplementary Figure 2a). Moreover, AMG517-treated mice showed a markedly increased edema when compared with vehicle- or A967079-treated mice (Supplementary Figure 2b). These results are consistent with results from genetic ablation studies and have further confirmed the distinct roles of TRP channels in SADBE-induced skin inflammation and persistent itch.

3. Scratching was not totally eliminated in the *Trpa1*^{-/-}/*Trpv1*^{-/-} mice (Lines 211 – 218). There was still a substantial response. What is the mechanistic basis for this residual pruritic effect – is it related to other actions on sensory neurons, perhaps increased action potential firing due to enhanced voltage gated sodium currents (Qu et al 2014;) or reduced potassium currents?

Response: Thanks for bringing this up. To address this important question we have performed additional experiments and compared the number of scratches of the *Trpa1*^{-/-}/*Trpv1*^{-/-} dKO mice injected with either vehicle or SADBE. Although the SADBE group tended to scratch more when compared with vehicle group, there was no significant difference between these two groups (Figure 2). Moreover, even 10 mM SADBE could not induce action potential firings in DRG neurons dissociated from the *Trpa1*^{-/-}/*Trpv1*^{-/-} dKO mice (please also see our the response to your Comment 9), we thus conclude that both TRPA1 and TRPV1 are required for SADBE-induced cellular and behavioral responses.

Figure 2. The number of scratching bouts induced by intradermal injections of vehicle or SADBE in *Trpa1^{-/-}/Trpv1^{-/-}* dKO mice. n.s, not significant. Student's *t*-test.

4. The mechanism by which the absence of either TRPV1 expressing sensory neurons (RTX-treatment) or functional TRPV1 channels (*Trpv1^{-/-}* mice) increases SADBE-evoked inflammation is not investigated in depth. Other studies have shown opposite effects of RTX treatment on inflammation elicited by some haptens (e.g. Imiquimod – Riol-Blanco et al 2014 Nature 510: 157) and similar effects of TRPV1 inhibition for other haptens (e.g. Oxazalone - Bánvölgyi A et al 2005 J Neuroimmunol 169: 86). There is a similar dichotomy between the conclusion that TRPA1 does not influence inflammation in this SADBE model but is important for oxazalone-evoked inflammation (Liu et al., 2013 FASAB J 27:3549). The authors provide some evidence for cytokine changes to explain their findings but the mechanisms by which sensory neurons drive these changes and why this is linked to TRPV1 and not to TRPA1 are not addressed.

I recommend that the authors add more details or amend wording to aid the reader.

Response: Thanks for your constructive suggestions. We have performed additional experiments and provided evidence for the involvement of dermal macrophages in TRPV1-mediated modulation of the SADBE-induced skin inflammation in the revised manuscript.

5. Methods. Lines 439 – 453, 459 - 462 and e.g. Figure 1. A). The protocols for administration of SADBE need to be consistently explained in the methods and the figures. The diagrams in the figures show SADBE administered first to the abdomen and then to the ear. But the scratching behavior is measured after SADBE is administered to either the nape of the neck. B) The timing of SADBE administration when measuring ear edema is unclear. The method (line 461) states measurements made after the challenge at day 3, but the protocol shown in figure 1 shows measurement 3 days after the last SADBE treatment. Do you mean ‘measurements were made 3 days after the last SADBE challenge?’

Response: A) Thanks for pointing out this important issue. We apologize for not making this clear in our original submission. Mice were all challenged to their ears in our experiments. We have corrected this description in the revised manuscript. B) Sorry for the confusion. As shown in the schematic experimental protocol, we did the measurements 3 days after the last SADBE challenge. We have refined our statement in the revised manuscript.

6. What vehicle controls were used in the *in vitro* experiments and in the intradermal injection experiments? Was it acetone? If so, what was the final acetone concentration and did this evoke scratching. Vehicle controls should be presented.

Response: Sorry for the missing this important information. For both *in vitro* experiments and intradermal injections, a 2.3M stock solution was made by dissolving SADBE in DMSO with a ratio of 1:1 (by volume). To prepare the working solution, this stock was further diluted in saline. We have added related descriptions in the methods section in the revised manuscript. For the behavior testing, the final DMSO concentration in the vehicle was lower than 2%, which didn't not evoke any significant scratching or wiping behavior as we showed in the revised manuscript (Figure 5a).

7. Line 399 onwards - Methods. There are no methods given for the current clamp electrophysiology experiments. These should be added.

Response: Thanks for pointing this out. We have added the methods of electrophysiology in the revised manuscript.

8. Line 111. Was there any measurement to show that the FTY720 treatment was effective in blocking T cell migration?

Response: Thanks for bringing this up. As an immunosuppressant drug, FTY720 has been widely used to inhibit lymphocyte emigration from lymphoid organs. By downregulating Sphingosine-1-phosphate receptor 1 (S1P₁), the mechanism of FTY720-induced lymphocyte sequestration was reported by a Nature paper in 2004. [1]. So far, FTY720 is commonly used in immunity study of effector T cells and even proved by FDA in the treatment of T cell-mediated autoimmune diseases. By searching references, we also found that the effect of FTY720 is consistent among different groups [2-4] and some of these studies did not further confirm the effect of FTY720 in blocking T cell migration [5].

[1] Matloubian ME, et al. "Lymphocyte egress from thymus and peripheral lymphoid organs is dependent on S1P receptor 1." *Nature* 427.6972 (2004): 355-360.

[2] Benechet AP, et al. "T cell-intrinsic S1PR1 regulates endogenous effector T-cell egress dynamics from lymph nodes during infection." *Proceedings of the National Academy of Sciences* 113.8 (2016): 2182-2187.

[3] Schenkel JM, Masopust D. "Tissue-resident memory T cells." *Immunity* 41.6 (2014): 886-897.

[4] Hondowicz BD, et al. "Interleukin-2-dependent allergen-specific tissue-resident memory cells drive asthma." *Immunity* 44.1 (2016): 155-166.

[5] Riol-Blanco L, et al. "Nociceptive sensory neurons drive interleukin-23-mediated psoriasiform skin inflammation." *Nature* 510.7503 (2014): 157-161.

9. Lines 155 – 164. The results shown in Figure 3, panels a-d need to be expressed quantitatively. How many neurons were studied, how many neurons responded to SADBE, how many coverslips? What were the percentages - the term 'similar percentages' is vague? The 3mM SADBA concentration used in these DRG experiments is a sub-maximally effective concentration for TRPV1 activation in the heterologous expression experiments (see Figure 4g). From Fig 3b it looks as though there are many TRPV1 expressing neurons that did not respond to SADBE. Were there SADBE responsive neurons that were not AITC or capsaicin sensitive in DRG neurons from wild-type, *Trpa1*^{-/-} or *Trpv1*^{-/-} mice? What was the percentage of responding neurons in the double knockout (*Trpa1*^{-/-}/*Trpv1*^{-/-}) DRG neurons – was it really 0%?

Response: Thanks for your suggestions. To address your questions, we have performed additional experiments and also added detailed description for Figure 3 in the revised manuscript.

As suggested, we tried higher concentration of SADBE (10 mM) in both patch-clamp recordings and calcium imaging assays. Robust depolarization and spontaneous action potential firings were elicited by 10 mM SADBE in current clamp recordings, which desensitized the DRG neurons isolated from *wt* mice and blunted the AITC and capsaicin responses in the same cells (Figure 3a and 3b). Similar results were also found in calcium imaging experiments (Figure 3e), i.e. neither AITC nor capsaicin evoked significant calcium response after application of 10 mM SADBE. Furthermore, we compared the percentage of *wt* DRG neurons responded to 3 mM and 10 mM SADBE. Although there was a trend of increase in the number of neurons activated by 10 mM SADBE, there was no significant difference between the 3 mM and 10 mM groups (Figure 3g). Interestingly, a subpopulation of capsaicin-sensitive DRG neurons showed tiny or no calcium influx in response to 3 mM SADBE in calcium imaging assays (see Figure 3b in the revised manuscript), which might be due to its weaker potency to activate TRPV1.

To further confirm no other receptors other than TRPA1 and TRPV1 were involved in SADBE activation of sensory neurons, we used 10 mM SADBE to fully activate DRG neurons. As shown in Figure 3c, 3d and 3f below, no membrane depolarization, action potential firing or calcium influx was found in both current-clamp recordings and calcium imaging assays in response to 10 mM SADBE in DRG neurons isolated from *Trpa1*^{-/-}/*Trpv1*^{-/-} *dKO* mice, suggesting that TRPA1 and TRPV1 are the only SADBE receptors in mouse DRG neurons.

Figure 3. 10 mM SADBE did not activate DRG neurons isolated from *Trpa1^{-/-}/Trpv1^{-/-}* dKO mice. (a) SADBE-induced depolarization of membrane potential and action potential firing in DRG neurons isolated from *wt* mice; (b) Quantification of depolarization of membrane potentials induced by SADBE, AITC and Cap. $n=5$, *** $p<0.001$, **** $p<0.0001$, ANOVA; please note that AITC- and Cap-activated responses were markedly reduced when applied after SADBE. (c) SADBE did not induce depolarization of membrane potential and action potential firing in DRG neurons isolated from *Trpa1^{-/-}/Trpv1^{-/-}* dKO mice; ATP was used as a positive control; (d) Quantification of depolarization of membrane potentials induced by SADBE, AITC, Cap and ATP. $n=5$, **** $p<0.0001$, ANOVA; (e-f) SADBE-induced calcium influx in *wt* ($n=5$ coverslips, 689 neurons) and *Trpa1^{-/-}/Trpv1^{-/-}* dKO ($n=5$ coverslips, 752 neurons) DRG neurons; (g) Percentages of *wt* DRG neurons responded to 3 mM and 10 mM SADBE. n.s, not significant. Student's *t*-test.

10. Figure 3, panels b and c. The time courses of the responses in the *Trpa1^{-/-}* and *Trpv1^{-/-}* mouse DRG neurons looks very different (*Trpa1^{-/-}* transient, *Trpv1^{-/-}* more persistent). Was this a consistent finding? Two components of response time courses also appear to be present in the wild-type DRG neurons. A difference in time course would be consistent with the time courses shown for TRPV1- and TRPA1-mediated calcium responses in Figure 4a and b.

Response: Thanks for bringing up this interesting phenomenon. We did observe two different patterns of calcium responses mediated by TRPA1 or TRPV1 channels. TRPA1 channel is continuously activated while TRPV1 is quickly desensitized upon activation in calcium imaging experiments, which might contribute to the weaker potency of SADBE activation of TRPV1.

Unfortunately, based on these results we can't further interpret whether these two patterns lead to any behavior differences.

11. Lines 191 – 194 & Figure 4. The SADBE-evoked responses of hTRPA1 mutants are depressed, but is this due to a reduced expression level? For human TRPA1 3C mutants, responses to a non-electrophilic agonist such as carvacrol would demonstrate that the channels were well expressed.

Response: Thanks for your suggestions. (1) As reported by other groups [1-2], both three cysteine and lysine residues are important for activation of TRPA1 by reactive compounds through covalent modification, which can profoundly affect channel gating and channel conformation. Thus, these mutants might show depressed or no response to different reactive compounds when compared with native TRPA1 channel. However, the expression patterns are similar among cysteine mutants and wildtype TRPA1 [1]. In our case, three cysteines play a major role in mediating sensitivity to SADBE while TRPA1-K also showed a 4.6-fold reduction in EC_{50} . These factors may contribute to the depressed response to SADBE in TRPA1 mutants.

(2) As suggested, we used flufenamic acid (FFA) as a positive control in patch clamp recording as our previous studies showed that FFA is a non-electrophilic agonist of TRPA1 [3]. Consistent with our previous results, SADBE barely evoked inward- and outward-currents while FFA strongly activated TRPA1-3C channel (Figure 4), confirming the transfection efficiency.

Figure 4. Whole-cell membrane currents elicited by SADBE in HEK293 cells transfected with TRPA1-3C construct. (a) Representative I-V curves of TRPA1-3C currents in response to 3 mM SADBE and 100 μ M FFA; (b) Quantification of SADBE- and FFA-induced TRPA1-3C currents. *** $p < 0.001$, Student's t -test.

[1] Macpherson LJ, et al. "Noxious compounds activate TRPA1 ion channels through covalent modification of cysteines." *Nature* 445.7127 (2007): 541-545.

[2] Hinman A, et al. "TRP channel activation by reversible covalent modification." *Proceedings of the National Academy of Sciences* 103.51 (2006): 19564-19568.

[3] Hu H, et al. "Zinc activates damage-sensing TRPA1 ion channels." *Nature Chemical Biology* 5.3 (2009): 183-190.

12. Lines 197 – 198 and Figure 4. The concentration response curve for the TRPV1 M548L mutant appears to be shifted leftwards compared to the wild-type TRPV1, yet the authors say there was no effect. Some quantification of results and evidence of reproducibility should be added to justify the conclusion.

Response: Sorry for the confusion. We agree with this reviewer that the potency of SADBE on M548L was significantly increased when compared with *wt* (Supplementary Table 2 in revised manuscript), which may be due to the enhanced interaction between SADBE and Leucine. In the original manuscript, we meant that SADBE potency was not decreased by substitution of R115 and M548. We have corrected the description and added the quantification results in the revised manuscript (Supplementary Table 1 and Table 2 in the revised manuscript).

13. Lines 209 - 210. Presumably there are some words missing in this sentence (e.g. mice lacking functional TRPA1 and/or TRPV1).

Response: Thanks. We have revised this sentence according to your suggestion.

Reviewer #2 (Remarks to the Author):

This study shows distinct contributions of TRP channels to skin inflammation and itch in the SADBE-induced contact dermatitis model. By using TRPA1 knockout (KO), TRPV1 KO and TRPA1/TRPV1 double KO mice, the authors demonstrate that persistent scratching caused by SADBE treatment is mediated through both TRPA1 and TRPV1 channels. Calcium imaging and electrophysiological experiments using acutely dissociated DRG cells and HEK cells transfected with TRPA1 or TRPV1 show that SADBE can activate directly both TRP channels. Unlike persistent scratching induced by repeated application of SADBE, acute scratching induced by intradermal SADBE injection is mediated mainly through TRPA1. In contrast to scratching behavior, deficiency or pharmacological inhibition of TRPV1 exacerbates skin inflammation probably through promoting Th1 cytokines. These findings strongly suggest that SADBE causes itch independent of skin inflammation. Although most experiments were appropriately conducted and the manuscript is well organized, there remain a number of questions that should be addressed.

1. The authors found an increase in scratching at 3 days after the final SADBE challenge in sensitized mice (Fig. 1), thereby regarding this scratching as “persistent” itch. On the other hand, they show that intradermal injection of SADBE elicited “acute” scratching in naïve mice (Fig. 5). I strongly recommend that the authors should present the time course of scratching after SADBE application in both the “persistent” and “acute” model, and also should examine the effects of TRP channel deficiency on scratching at different time points. Perhaps, increased scratching will be observed immediately after the final application of SADBE in sensitized mice; does the

deficiency of TRPA1 alone or both TRPA1 and TRPV1 inhibit this scratching response? Also, how long does the scratching induced by SADBE injection in naïve mice continue? The present results raise an interesting question whether persistent scratching is due to continuation of direct activation of TRPA1 and/or TRPV1 by SADBE existing in the skin, or whether other mediator(s) induced by repeated SADBE application eventually activate TRPA1 and TRPV1 channels.

Response: Thanks for your valuable suggestions. As recommended, we have recorded the time course of SADBE-induced acute and persistent itch in *wt* and mice deficient in TRPA1, TRPV1, and both TRPA1 and TRPV1.

In the acute itch model, scratching behavior was immediately observed after intradermal injection of SADBE and gradually diminished 30 min later in *wt* mice. As expected, *Trpa1*^{-/-} and dKO but not *Trpv1*^{-/-} mice showed markedly reduced scratching behavior in response to SADBE injection, suggesting that TRPA1 is required in SADBE-induced acute itch (Figure 5a).

In the model of SADBE-induced persistent itch, as the reviewer envisioned, *wt* mice showed a robust scratching response within the first hour after SADBE application presumably due to direct activation of TRPA1/V1 channels by SADBE. Although the number of scratching bouts was significantly reduced by more than half, which may due to the potential TRP channel desensitization and hapten clearance by immune response, itch sensation remained steady during the first 24 hours and dramatically increased at day 2 and day3 (Figure 5b). Interestingly, we also found that other mediators (such as 5-HT) released by activated innate immune cells contribute to the increased itch sensation in the later phase of SADBE-induced CHS (manuscript accepted by *J. Allergy Clin. Immunol.*).

Taken together, SADBE could induce persistent itch sensation by directly activating TRP channels and/or indirectly inducing release of endogenous pruritogens from innate immune cells. Nevertheless, TRPA1 and TRPV1 are the dominant downstream targets in both pathways as genetic ablation of TRPA1/TRPV1 significantly reduced SADBE-induced persistent itch sensation.

Figure 5. Time courses of SADBE-induced acute and persistent itch. (a) Time course of acute itch responses elicited by intradermal injection of 30mM SADBE in the neck of *wt*, *Trpa1*^{-/-}, *Trpv1*^{-/-} and *Trpa1*^{-/-}/*Trpv1*^{-/-} dKO mice. * $p<0.05$; ** $p<0.01$, *** $p<0.001$, ANOVA; (b) Time course of persistent itch responses elicited by SADBE using the standard sensitization/excitation protocol as described in the revised manuscript. Mice were recorded from 0-1, 2-3, 4-5, 24, 48 and 72 hour after the final challenge of SADBE. * $p<0.05$; ** $p<0.01$, *** $p<0.001$, **** $p<0.0001$, ANOVA.

2. As shown in Figs. 1e and i, SADBE challenge on the nape of the neck causes scratching, irrespective of systemic presensitization on the abdominal skin, although skin inflammation is significantly but very weakly promoted by the presensitization. Therefore, SADBE causing symptoms, especially itch (or scratching), should be considered as irritant contact dermatitis (ICD), but not allergic contact dermatitis (ACD).

Response: We strongly agree with this reviewer and added new paragraphs in the discussion. In SADBE-induced CHS, both allergic and irritant contact dermatitis could occur but persistent itch is a typical irritant contact dermatitis response. On the other hand, we could not rule out the possibility that endogenous pruritogens including many cytokines released in the immune-mediated inflammation process could also contribute to the persistent itch.

3. The authors showed that intradermal SADBE injection in naïve mice exclusively caused scratching compared to wiping behaviors (Fig. 5); however, each behavior should be compared to when the vehicle was injected. Scratching and wiping behaviors should not be compared to each other. Furthermore, Qu et al. have reported that repeated SADBE application induces not only itch-related scratching but also pain-like behaviors, such as wiping and licking (Qu L, Brain, 2014). As described, both TRPA1 and TRPV1 play a critical role in pain. Have you checked pain behaviors in your SADBE model? How does the TRP channel deficiency affect pain caused by SADBE treatment?

Response: Thanks for your suggestions. We have modified Figure 5 in the revised manuscript. For the pain-related behavior, please also see our response to Reviewer #1's Comment 1.

We have performed additional experiments investigating SADBE-elicited pain-like behavior. Surprisingly, we didn't find obvious wiping behavior by injection 30 mM SADBE with cheek model as shown in the revised manuscript (Figure 5a). Furthermore, we didn't observe any paw licking or flicking behavior within the first 10 mins after intraplantar injection of 30 mM SADBE (data not shown). To further exclude the possibility that SADBE contributes to mechanical and thermal pain sensitivities, we have performed Von Frey and Hargreaves tests after paw injection of 30 mM SADBE. As shown in Fig 1a and 1b (response to Reviewer #1's Comment 1), we did not find significant difference in paw withdrawal threshold to mechanical stimuli or paw withdrawal latency to heat stimuli compared with mice injected with vehicle only. These results suggested that SADBE acts primarily as an itch-evoking compound when applied acutely at the given dosage.

We further tested pain-like behavior in SADBE-induced allergic contact dermatitis model. As shown in Figure 1e and 1f (response to Reviewer #1's Comment 1), no wiping behavior was observed while scratching behavior gradually increased during the first 3 days when compared with the vehicle control group. However, excessive scratching produced excessive skin lesions at day 4 (Figure 1c and 1d, response to Reviewer #1's Comment 1) where scratching behavior was markedly decreased while wiping behavior began to show up, suggesting that tissue damage and injury promotes pain response but suppresses itch response. This finding is consistent with previous findings that pain sensation constitutively suppresses itch sensation [1,2].

We used 20 μ l of 0.5% SADBE compared with 25 μ l of 1% SADBE used by Qu et al. in their original papers. Our protocol yielded a much better itch response because the low concentration of SADBE we used delayed the development of pain behavior caused by excessive scratching and prolonged the chronic itch phase. Taken these together, we suggest that SADBE elicits neither acute pain nor chronic pain sensation directly in *wt* mice at the concentrations we used in our studies.

[1] Liu Y, et al. "VGLUT2-dependent glutamate release from nociceptors is required to sense pain and suppress itch." *Neuron* 68.3 (2010): 543-556.

[2] Lagerström MC, et al. "VGLUT2-dependent sensory neurons in the TRPV1 population regulate pain and itch." *Neuron* 68.3 (2010): 529-542.

4. Regarding statistics: (1) when comparing among wild type, TRPA1 KO, TRPV1 KO and double KO mice, the authors should compare all pairs of groups. Most of the comparisons were done only vs. the wild type group. (2) The significant differences in Fig. 3j seem to be wrong. (3) The authors should perform statistical analysis in studies shown in Figs. 4g and h.

Response: Thanks for pointing this out. (1-2) We have performed additional statistical analysis and revised all related figures according to your suggestions; (3) We have added supplementary Table 1 and 2 with a complete statistical analysis.

5. How did you decide the concentration and dose of SADBE (in vitro: 3 mM; in vivo: 30 mM)? In addition, the authors should present the volume of SADBE solution injected intradermally and what the vehicle was.

Response: Thanks for your comments.

(1) 0.5% SADBE was used in the induction of SADBE-induced CHS. The molar concentration is about 23 mM. So we decided to use 30 mM SADBE for the intradermal injection. For *in vitro* experiments, we have tried different concentrations and found that 3 mM was the most appropriate concentration because it was sufficient to activate DRG neurons without producing a robust desensitization (please also see response to Reviewer #1's comment # 9).

(2) We have added the description in the revised method section (please also see our response to Reviewer #1's comment #6).

6. Calcium imaging and electrophysiological experiments clearly showed mutual compensatory effect in the DRG neurons isolated from TRPA1 KO or TRPV1 KO mice (Fig. 3). On the other hand, SADBE-induced persistent scratching could be suppressed in either TRPA1 KO or TRPV1 KO mice (Fig. 2). If DRG neurons play a dominant role in SADBE-induced persistent itch as shown in Fig. 8, these results would be theoretically inconsistent. Is it possible that TRPA1- and/or TRPV1-expressed in other cells besides DRG neurons is involved in the persistent itch? The authors should fully address this question.

Response: Thanks for bringing up this question. Functional interaction between TRPV1 and TRPA1 occur in several ways [1] and accumulating evidence suggests TRPA1 and TRPV1 assemble channel complexes in heterologous expression systems and sensory neurons [2-4]. Weng et al also proposed a TRPA1-TRPV1 complex model and revealed how this complex contributed to persistent pain sensation under the control of Tmem100 [5]. In brief, TRPA1 activity could be potentiated by TRPV1 in the presence of Tmem100. However, TRPA1- or TRPV1-mediated currents were partially inhibited by Tmem100 in DRG neurons isolated from *Trpv1*^{-/-} or *Trpa1*^{-/-} mice, respectively. This theory could also be applied to our findings in TRPA1- and TRPV1-mediated persistent itch. Although TRPA1 or TRPV1 still mediated calcium influx in *Trpv1*^{-/-} or *Trpa1*^{-/-} mice, we did observe reduced amplitude in calcium responses in DRG neurons from *Trpa1*^{-/-} and *Trpv1*^{-/-} mice (Fig 3a versus Fig 3b and 3c in the revised manuscript). The compromised channel function may lead to the reduced scratching in the TRP channel deficient mice.

It remains controversial whether TRPA1 and TRPV1 channels are expressed in nonneuronal cells [6-10]. To exclude the possibility that functional TRP channels expressed by keratinocytes and immune cells contribute to SADBE-induced itch, we performed calcium imaging experiments with skin-resident cells dissociated from the ear preparations of *wt* mice. Neither AITC nor capsaicin evoked measurable calcium responses (Fig 6), suggesting it is unlikely that the SABDE-induced persistent itch is mediated by TRPA1 and TRPV1 channels that are expressed by nonneuronal cells.

Fig 6. Representative traces showing AITC- and capsaicin-evoked $[Ca^{2+}]_i$ response in skin-resident cells freshly isolated from ear preparations of *wt* mice. Ionomycin is used as a positive control. AITC 100 μ M, capsaicin 300 nM, ionomycin 1 μ M. n=5 independent repeats.

- [1] Julius, David. "TRP channels and pain." *Annual review of cell and developmental biology* 29 (2013): 355-384.
- [2] Akopian, Armen N., et al. "Transient receptor potential TRPA1 channel desensitization in sensory neurons is agonist dependent and regulated by TRPV1-directed internalization." *The Journal of physiology* 583.1 (2007): 175-193.
- [3] Fischer, Michael JM, et al. "Direct evidence for functional TRPV1/TRPA1 heteromers." *Pflügers Archiv-European Journal of Physiology* 466.12 (2014): 2229-2241.
- [4] Staruschenko, Alexander, Nathaniel A. Jeske, and Armen N. Akopian. "Contribution of TRPV1-TRPA1 interaction to the single channel properties of the TRPA1 channel." *Journal of biological chemistry* 285.20 (2010): 15167-15177.
- [5] Weng, Hao-Jui, et al. "Tmem100 is a regulator of TRPA1-TRPV1 complex and contributes to persistent pain." *Neuron* 85.4 (2015): 833-846.
- [6] Fernandes, E. S., M. A. Fernandes, and J. E. Keeble. "The functions of TRPA1 and TRPV1: moving away from sensory nerves." *British journal of pharmacology* 166.2 (2012): 510-521.
- [7] Bertin, Samuel, et al. "The ion channel TRPV1 regulates the activation and proinflammatory properties of CD4+ T cells." *Nature immunology* 15.11 (2014): 1055-1063.
- [8] Zappia, Katherine J., et al. "Mechanosensory and atp release deficits following keratin14-cre-mediated TRPA1 deletion despite absence of TRPA1 in murine keratinocytes." *PloS one* 11.3 (2016): e0151602.
- [9] Himi, N., et al. "Calcium influx through the TRPV1 channel of endothelial cells (ECs) correlates with a stronger adhesion between monocytes and ECs." *Advances in medical sciences* 57.2 (2012): 224-229.
- [10] LaMotte, Robert H., Xinzhong Dong, and Matthias Ringkamp. "Sensory neurons and circuits mediating itch." *Nature reviews Neuroscience* 15.1 (2014): 19-31.

7. The responses to SADBE in TRPV1-expressing HEK293 cells were clearly weaker than those in TRPA1-expressing cells (Fig. 4). Indeed, the EC₅₀ value for TRPV1-expressing cells was 5.6 times higher than that for TRPA1 (1.30 mM for TRPA1 vs. 7.26 mM for TRPV1). Why is that? Also, what concentration of SADBE did you use? If it was 3 mM similar to other *in vitro* experiments, the authors should use a higher (or submaximal) concentration (e.g., 10 mM) to determine the SADBE activation of TRPV1.

Response: Although TRPV1 and TRPA1 channels are structurally related, non-selective cation channels; these two channels possess different primary sequences and properties. Therefore, it is not surprising that they show different affinities against different chemicals. For instance, capsaicin and AITC selectively activate TRPV1 and TRPA1, respectively. In our case, SADBE possesses different potencies to activate TRPA1 and TRPV1, which might result from distinct intermolecular force and/or interaction mode caused by special residues between these two channels, for instance, cysteine and lysine residues in TRPA1 and tyrosine, serine, and threonine residues in TRPV1.

For *in vitro* experiments, 3 mM SADBE was used to activate TRPV1 because a higher concentration of SADBE (10 mM) excessively activated and subsequently desensitized DRG neurons in both calcium imaging assays and patch clamp recordings, which blunts AITC and

capsaicin responses in the same cells (please also see Fig 3 in response to Reviewer #1's comment # 9). Therefore, we used 3mM but not 10 mM SABDE to classify DRG neurons together with capsaicin and AITC.

8. The authors suggest that increased Th1 cytokines would contribute to aggravation of SADBE-induced skin inflammation by TRPV1 deficiency. On the other hand, SADBE-induced skin inflammation was not affected in Rag1^{-/-} mice, which lack T cells (Fig. 1f), suggesting that T cells are hardly required for SADBE-induced skin inflammation. Also, since the authors have not examined the effect of NK cell depletion alone on skin inflammation, the role of NK cells and T cells in skin edema by SADBE remains unclear. They thus should further address this question and revise the working model (Fig. 8) to fit their findings.

Response: Thanks for your suggestions. We have tested the function of NK cells in SADBE-induced inflammation by depletion of NK cells using anti-NK1.1 antibodies and we found that depletion of NK cells alone didn't alleviate SADBE-induced skin edema (Fig 1g in the revised manuscript), which is consistent with a previous report [1]. On the other hand, *Rag-1*^{-/-} mice receiving anti-NK1.1 antibody displayed a significantly decreased skin edema in response to SADBE challenges (Fig 1h in the revised manuscript), suggesting neither T/B cell nor NK cell priming alone is sufficient to mediate SABDE-induced inflammation but deficiency in all lymphocytes is effective in reducing SADBE-induced skin inflammation. We have already modified our working model according to these new findings.

[1] O'Leary, Jacqueline G., et al. "T cell-and B cell-independent adaptive immunity mediated by natural killer cells." *Nature immunology* 7.5 (2006): 507.

Reviewer #3 (Remarks to the Author):

Jing et al. showed that the SADBE-induced persistent itch was not depend on lymphocytes, but mediated by TRPA1 and TRPV1 channels. They demonstrated SADBE can directly activate both TRPA1 and TRPV1 in vivo using freshly isolated DRG cells and TRPA/TRPV1-expressing HEK293 cells. Further, their observation suggests that TRPV1 also affect SADBE-induced ear swelling via inhibiting the production of Th1 cytokines. In this paper, data were well-presented; however, I had concerns relating to the interpretation of some of the results, and lack of mechanistic insight. Followings are my specific comments:

1. Fig.1e showed that irritant response (innate response) by SADBE challenge was very strong with this experimental protocol. It makes hard to evaluate adaptive immune response in this condition. Author need to modify the protocol to reduce irritant response to SADBE.

Response: Thanks for your suggestions. In this manuscript, we aimed to determine the distinct roles of TRP channels in persistent itch and inflammation. As we showed in Figure 1e and 1i in

the manuscript, SADEB challenges led to a significantly increased inflammatory response that was mediated by cutaneous immune cells.

On the other hand, SADBE-induced persistent itch is mainly an irritant response that is mediated by TRP channels. Moreover, TRPV1 modulates skin inflammation. We have tried three different concentrations of SADBE (0.25%, 0.5%, and 1%) in order to find the best to generate a moderate contact dermatitis that we can use to investigate the mechanisms of skin inflammation (mediated by immune cells) and persistent itch (mediated by TRP channels).

We found that 0.25% SADBE was unable to induce an irritant contact dermatitis (Fig 7) while 1% SADBE-induced response was too strong to investigate the function of TRP channels (immune cells were excessively activated and TRP channels were desensitized). On the other hand, 0.5% SADBE produced a moderate skin inflammation associated with persistent itch which we used in our studies to effectively determine the roles of TRP channels in SADBE-induced CHS.

Fig 7. 0.25% SADBE did not elicit contact hypersensitivity in *wt* mice. Skin edema (a) and scratching number (b) of the mice treated with 0.25% SADBE was not significantly increased when compared with the vehicle treated mice. n=5 per group. n.s., not significant. Student's *t*-test.

2. Authors need to explain why ear swelling was not attenuated in Rag1-deficient (Fig.1f) and FTY720-treated (Fig.1g) mice compared to WT controls. It seems that authors just failed to induce adaptive immune response in these experiments.

Response: Thanks for bringing this important point. The induction of CHS was effective as these mice displayed significantly increased inflammation and spontaneous scratching (see Figure 1f and 1g in the revised manuscript). In a Nature Immunology paper [1], the authors also showed an intact cutaneous CHS in Rag2-deficient mice and suggested that NK cells are both necessary and sufficient to mediate a potent CHS response in the absence of other adaptive lymphocytes. We have also tried to induce CHS response in *wt* mice treated with anti-NK1.1 antibodies or control IgG antibodies. NK cell depletion alone did not abolish CHS response in *wt* mice, which can still develop a T-cell dependent CHS. On the other hand, Rag1-deficient mice treated with anti-NK1.1 antibodies had a reduced inflammation (Figure 1h in revised manuscript). Taken together,

both T/B cells and NK cells are likely involved in mediating the antigen-specific adaptive recall response after SADBE challenge.

[1] O'Leary, Jacqueline G., et al. "T cell-and B cell-independent adaptive immunity mediated by natural killer cells." *Nature immunology* 7.5 (2006): 507.

3. Authors need to explain the interpretation of the result shown in Fig.1f. Does this result suggest SADBE-induced CHS response is mediated by NK cells but not by T/B cells? If so, is there an antigen-specificity in this response?

Response: Sorry for the confusion. We have made changes in Fig 1 of the revised manuscript to emphasize the functions of T/ B cell and NK cell in the revised manuscript.

4. In Fig.1e, ear thickness increment looks over than 100% in SADBE-sensitized and SADBE-challenged group; however, the increment in same group looks less than 80% in Fig.1f and Fig.1g. Moreover, it was less than 50% in Fig. 6b. What causes these discrepancies?

Response: Thanks for bringing up this question. We used mice of 8-weeks of age with a body weight of 20-23g in our behavior testing. However, the *Trpa1^{-/-}/Trpv1^{-/-}* dKO mice were lighter at 8-weeks of age when compared with *wt*, *Trpa1^{-/-}*, and *Trpv1^{-/-}* mice. We thus used the *Trpa1^{-/-}/Trpv1^{-/-}* dKO mice at 12 weeks of age when they had a comparable body weight. Similarly, the *Rag1*-deficient mice were used at about 10 weeks of age. Although we do not understand the mechanisms underlying the relationship between age and immunity, it seems the severity of inflammation decreased as the mice matured. On the other hand, these discrepancies of the inflammation at different ages shouldn't affect our conclusions as mice were strictly matched with their littermate control groups in our experiments.

Figure 8. Body weight of *wt*, *Trpa1^{-/-}*, *Trpv1^{-/-}* and *Trpa1^{-/-}/Trpv1^{-/-}* dKO mice at 8 weeks (a) and 12 weeks (b) old. Statistical significance was indicated by asterisk. * $p < 0.05$; *** $p < 0.001$, ANOVA.

5. Authors demonstrated that SADBE can directly activate TRPA1/TRPV1 channels. However, they did not present any data evaluating the indirect effect of SADBE; for instance, keratinocytes,

mast cells, ILCs, which can be activated by SADBE might subsequently activate TRP channels, as authors described.

Response: Thanks for bringing up this important point. In this manuscript, we mainly focused on the distinct roles of TRP channels in SADBE-induced inflammation and persistent itch. The classic hapten-induced inflammatory response mediated by adaptive immune system has already been well studied. Moreover, we did not find functional expression of TRPA1/TRPV1 by keratinocytes and innate immune cells in the skin (see also Fig 6 in the response to Review #2's Comment 6). Therefore, only the functions of TRPA1/TRPV1 in primary sensory neurons were included in this manuscript. However, we do not exclude the possibility that other TRP channel-expressed resident skin cells might be involved in SADBE-induced inflammatory response and persistent itch.

6. TRPA1 and TRPV1 play roles in SADBE-induced scratching behavior and ear swelling via Th1 cytokine production, respectively; although in-vivo data suggest that their function in calcium influx is compensable in response to SADBE. Authors need to discuss the mechanistic insight of this discrepancy.

Response: Thanks for your suggestions. Functional interaction between TRPV1 and TRPA1 occur in several ways [1] and accumulating evidence suggests TRPA1 and TRPV1 assemble channel complexes in heterologous expression systems and sensory neurons [2-4]. Weng et al also proposed a TRPA1-TRPV1 complex model and revealed how this complex contributed to persistent pain sensation under the control of Tmem100 [5]. In brief, TRPA1 activity could be potentiated by TRPV1 in the presence of Tmem100. However, TRPA1- or TRPV1-mediated currents were partially inhibited by Tmem100 in DRG neurons isolated from *Trpv1*^{-/-} or *Trpa1*^{-/-} mice, respectively. This theory could also be applied to our findings in TRPA1- and TRPV1-mediated persistent itch. Although TRPA1 or TRPV1 still mediated calcium influx in *Trpv1*^{-/-} or *Trpa1*^{-/-} mice, we did observe reduced amplitude in calcium responses in DRG neurons from *Trpa1*^{-/-} and *Trpv1*^{-/-} mice (Fig 3a versus Fig 3b and 3c in the revised manuscript). The compromised channel function may lead to the reduced scratching in the TRP channel deficient mice.

It remains controversial whether TRPA1 and TRPV1 channels are expressed in nonneuronal cells [6-10]. To exclude the possibility that functional TRP channels expressed by keratinocytes and immune cells contribute to SADBE-induced itch, we performed calcium imaging experiments with skin-resident cells dissociated from the ear preparations of *wt* mice. Neither AITC nor capsaicin evoked measurable calcium responses (Fig 6), suggesting it is unlikely that the SADBE-induced persistent itch is mediated by TRPA1 and TRPV1 channels that are expressed by nonneuronal cells.

Fig 6. Representative traces showing AITC- and capsaicin-evoked $[Ca^{2+}]_i$ response in skin-resident cells freshly isolated from ear preparations of *wt* mice. Ionomycin is used as a positive control. AITC 100 μ M, capsaicin 300 nM, ionomycin 1 μ M. $n=5$ independent repeats.

- [1] Julius, David. "TRP channels and pain." *Annual review of cell and developmental biology* 29 (2013): 355-384.
- [2] Akopian, Armen N., et al. "Transient receptor potential TRPA1 channel desensitization in sensory neurons is agonist dependent and regulated by TRPV1-directed internalization." *The Journal of physiology* 583.1 (2007): 175-193.
- [3] Fischer, Michael JM, et al. "Direct evidence for functional TRPV1/TRPA1 heteromers." *Pflügers Archiv-European Journal of Physiology* 466.12 (2014): 2229-2241.
- [4] Staruschenko, Alexander, Nathaniel A. Jeske, and Armen N. Akopian. "Contribution of TRPV1-TRPA1 interaction to the single channel properties of the TRPA1 channel." *Journal of biological chemistry* 285.20 (2010): 15167-15177.
- [5] Weng, Hao-Jui, et al. "Tmem100 is a regulator of TRPA1-TRPV1 complex and contributes to persistent pain." *Neuron* 85.4 (2015): 833-846.
- [6] Fernandes, E. S., M. A. Fernandes, and J. E. Keeble. "The functions of TRPA1 and TRPV1: moving away from sensory nerves." *British journal of pharmacology* 166.2 (2012): 510-521.
- [7] Bertin, Samuel, et al. "The ion channel TRPV1 regulates the activation and proinflammatory properties of CD4+ T cells." *Nature immunology* 15.11 (2014): 1055-1063.
- [8] Zappia, Katherine J., et al. "Mechanosensory and atp release deficits following keratin14-cre-mediated TRPA1 deletion despite absence of TRPA1 in murine keratinocytes." *PloS one* 11.3 (2016): e0151602.
- [9] Himi, N., et al. "Calcium influx through the TRPV1 channel of endothelial cells (ECs) correlates with a stronger adhesion between monocytes and ECs." *Advances in medical sciences* 57.2 (2012): 224-229.
- [10] LaMotte, Robert H., Xinzhong Dong, and Matthias Ringkamp. "Sensory neurons and circuits mediating itch." *Nature reviews Neuroscience* 15.1 (2014): 19-31.

7. Fig7: How about the expression level of IFN-gamma, the most important Th1 cytokine.

Response: Thanks for your suggestions. In the revised manuscript we have examined the expression level of IFN- γ in the ear preparations treated with SADBE in the *wt*, *Trpa1*^{-/-}, *Trpv1*^{-/-} and *Trpa1*^{-/-}/*Trpv1*^{-/-} dKO mice using Real-Time qRT-PCR. Unfortunately, it is unlikely that IFN- γ plays a dominant role in the enhanced inflammatory response in the TRPV1 deficient mice as the expression

level of IFN- γ was comparable among all groups tested (Supplementary Fig 5a in the revised manuscript).

8. Fig8: In this paper, there is no data that demonstrate the involvement of DDC in SADBE-induced CHS. Therefore, DDC function shown in Fig.8 seems overspeculation.

Response: Thanks for your comments. We have revised the working model in response to your comments.

REVIEWERS' COMMENTS:

Reviewer #1 (Remarks to the Author):

The revised manuscript by Feng and co-workers includes additional experiments that directly address some of my original comments and criticisms. They have also supplied some additional results in their response to reviewers' comments. The latter are welcome but I am surprised that this information was not included in the manuscript. Some of the comments that I originally made were questions that other readers may raise on close inspection of the data. I did not want the authors to personally convince me in their response but to add data that supported their case to the manuscript.

I therefore suggest that they:-

1. Add a comment about lack of paw licking or flicking behaviour after intraplantar injection of SADBE to the manuscript and add data on von Frey and Hargreaves tests in a Supplemental Figure.
2. Add data on the vehicle control (Figure 2 in their response to reviewers') to Figure 2d of the manuscript. This would show that the SADBE-evoked scratching was reduced to vehicle injection levels in the *Trpa1*^{-/-}/*Trpv1*^{-/-} mice.
3. Mention that DRG neuron responses evoked by 10mM SADBE were similar to those evoked by 3mM SADBE and similarly absent in the dKO DRG neurons. The lack of responses with a concentration of SADBE that strongly activates both TRPV1 and TRPA1 supports the argument that these TRP channels are the only SADBE 'receptor' in mouse DRG neurons.
4. Add the new information (response Figure 4) that flufenamic acid robustly activated TRPA1 lacking the 3 cysteines, but SADBE did not. This important control shows that the very small response to SADBE (Figure 4g in revised manuscript) is not due to a low expression level of TRPA1-3C. Details of the voltage used to calculate the data in response Figure 4 and the number of cells needs to be included.
5. Add a discussion on how they think that TRPV1 loss affects macrophages. This part of the new Figure 8 lacks any mechanistic insight. For example, are sensory neuropeptides such as substance P and CGRP likely to be involved?

Reviewer #2 (Remarks to the Author):

The authors have responded adequately to the previous review.

Reviewer #3 (Remarks to the Author):

none

REVIEWERS' COMMENTS:

Reviewer #1 (Remarks to the Author):

The revised manuscript by Feng and co-workers includes additional experiments that directly address some of my original comments and criticisms. They have also supplied some additional results in their response to reviewers' comments. The latter are welcome but I am surprised that this information was not included in the manuscript. Some of the comments that I originally made were questions that other readers may raise on close inspection of the data. I did not want the authors to personally convince me in their response but to add data that supported their case to the manuscript.

I therefore suggest that they:

1. Add a comment about lack of paw licking or flicking behaviour after intraplantar injection of SADBE to the manuscript and add data on von Frey and Hargreaves tests in a Supplemental Figure.

Response: Thanks for your suggestion. The data was added as Supplementary Figure 6.

2. Add data on the vehicle control (Figure 2 in their response to reviewers') to Figure 2d of the manuscript. This would show that the SADBE-evoked scratching was reduced to vehicle injection levels in the *Trpa1*^{-/-}/*Trpv1*^{-/-} mice.

Response: Thanks for your suggestion. We think the review may refer to Figure 5 as Figure 5 showed the acute injection of SADBE-induced itch behavior while Figure 2 showed the SADBE-induced chronic itch. As suggested, Figure 5 was revised and the scratching data from vehicle-injected mice was added.

3. Mention that DRG neuron responses evoked by 10mM SADBE were similar to those evoked by 3mM SADBE and similarly absent in the dKO DRG neurons. The lack of responses with a concentration of SADBE that strongly activates both TRPV1 and TRPA1 supports the argument that these TRP channels are the only SADBE 'receptor' in mouse DRG neurons.

Response: Thanks for your suggestion. The data were added as Supplementary Figure 3 and discussed in the related results.

4. Add the new information (response Figure 4) that flufenamic acid robustly activated TRPA1 lacking the 3 cysteines, but SADBE did not. This important control shows that the very small response to SADBE (Figure 4g in revised manuscript) is not due to a low expression level of TRPA1-3C. Details of the voltage used to calculate the data in response Figure 4 and the number of cells needs to be included.

Response: Thanks for your suggestion. The data were added as Supplementary Figure 6 and details were described in the figure legend.

5. Add a discussion on how they think that TRPV1 loss affects macrophages. This part of the new Figure 8 lacks any mechanistic insight. For example, are sensory neuropeptides such as substance P and CGRP likely to be involved?

Response: Thanks for your suggestion. We revised the discussion and the potential role of CGRP and SP in neuro-immune communication was discussed.

Reviewer #2 (Remarks to the Author):

The authors have responded adequately to the previous review.

Response: Thanks.

Reviewer #3 (Remarks to the Author):

None

Response: Thanks.